# CAN GRAPH FOUNDATION MODELS GENERALIZE OVER ARCHITECTURE?

**Benjamin Gutteridge**[1][*] **Michael Bronstein**[1,2] **& Xiaowen Dong**[1]
[1]University of Oxford, [2]AITHYRA

## ABSTRACT

Graph foundation models (GFMs) have recently attracted interest due to the promise of graph neural network (GNN) architectures that generalize zero-shot across graphs of arbitrary scales, feature dimensions, and domains. While existing work has demonstrated this ability empirically across diverse real-world benchmarks, these tasks share a crucial hidden limitation: they admit a narrow set of effective GNN architectures. In particular, current domain-agnostic GFMs rely on fixed architectural backbones, implicitly assuming that a single message-passing regime suffices across tasks. In this paper, we argue that *architecture adaptivity* is a necessary requirement for true GFMs. We show that existing approaches are non-robust to task-dependent architectural attributes and, as a case study, use *range* as a minimal and measurable axis along which this limitation becomes explicit. With theoretical analysis and controlled synthetic experiments, we demonstrate that fixed-backbone GFMs provably under-reach on tasks whose architectural requirements differ from those seen at training time. To address this issue, we introduce a framework that adapts effective GNN architecture at inference time by discovering and mixing task-specific linear graph operators, enabling zero-shot generalization across tasks with heterogeneous architectural requirements, without retraining. We validate our approach on arbitrary-range synthetic tasks and a suite of real-world benchmarks, demonstrating improved performance and robustness over existing domain-agnostic GFMs.[1]

## 1 INTRODUCTION

The remarkable success of foundation models in domains such as computer vision (Dosovitskiy et al., 2020; He et al., 2022; Radford et al., 2021; Wang et al., 2023c) and natural language processing (Devlin, 2018; Bubeck et al., 2023; Touvron et al., 2023) has sparked interest in graph foundation models (GFMs) as a means of achieving similar gains for generating and reasoning over graph-structured data. Several GFMs have been proposed towards this goal, but most of these are domain-specific, in that they are designed to handle multiple graph tasks on highly inter-related data (Wang et al., 2025), such as knowledge graphs, molecules or transport networks (Wu et al., 2025; Huang et al., 2024; Wang et al., 2023a; Galkin et al., 2023). Indeed, the lack of a shared 'vocabulary' among domains is one reason why effective *domain-agnostic* GFMs remain elusive. Nevertheless, several recent works have made progress in this direction; approaches include inference-time selection among domain experts (Xia & Huang, 2024), conversion of graphs to tabular format to leverage tabular foundation models (Eremeev et al., 2025; Hayler et al., 2025), and GNNs hybridized with large language models (LLMs) (Liu et al., 2025). The latter may use LLMs as feature encoders on backbone GNNs (Zhu et al., 2024), use GNNs as LLM 'tools' (Tang et al., 2024), or even use LLMs end-to-end on token or text representations of graphs (Perozzi et al., 2024).

In this paper, we focus our attention on works that tackle GFMs within the geometric learning paradigm, i.e., respecting the graph structure and symmetry principles offered by GNNs. Within this space, there have been two major contributions, both of which are focused on inductive node classification as a zero-shot 'inpainting' problem; i.e. given a graph and some reference node labels, classify the remaining nodes. The first, GraphAny (Zhao et al., 2024), can be interpreted as a

---

[*]Correspondence to `beng@robots.ox.ac.uk`
[1]Experiment code available at `https://github.com/BenGutteridge/GOBLIN`.

node-wise mixture-of-experts (MoE) across a fixed basis of linear GNNs. The second, Finkelshtein et al. (2025), builds on this, extending symmetry-aware architectures (Zaheer et al., 2017; Segol & Lipman, 2019) to formalize the 'triple symmetry' required for zero-shot GFM inference: node equivariance, label equivariance, and feature *in*variance.

Both GraphAny and triple-symmetry (TS-)GNNs are domain-agnostic, and both demonstrate their generalization on a set of real-world benchmarks that are diverse in graph size, topology, domain, task homophily/heterophily, feature dimension, and number of classes. However, in using these standard graph benchmarks to assess generalization capability, these methods end up strongly overfitting in a crucial area: backbone *GNN architecture*.

**Existing domain-agnostic GFMs assume a fixed architecture; generalization to tasks beyond the capabilities of that architecture is impossible without retraining.**

This implicit overfitting becomes clear when one considers the benchmarks used by these works. While they are diverse in certain ways (e.g., homophilic vs heterophilic), they are arguably all short-range tasks (Bamberger et al., 2025; Arnaiz-Rodriguez & Errica, 2025). Specifically, there is nothing to suggest that any of these tasks requires an approach more sophisticated than local message passing—and this is reflected in the underlying architectures: both GraphAny and TS-GNN are evaluated using MPNN backbones of only *two layers*.

In this paper, we argue that this limitation is not specific to the benchmarks considered, but is structural: fixed-backbone GFMs cannot, by construction, adapt to tasks whose architectural requirements differ from those implicitly encoded in their design. More broadly, we argue that *architecture adaptivity* is a necessary requirement for GFMs.

To make this limitation precise, we focus on *range* as a minimal and measurable architectural attribute. Range captures the distance (according to some metric on a graph, such as geodesic distance) over which node interactions must be aggregated in order to solve a task. While range is only one of many potential architectural axes along which GNNs may differ, it has the advantage of admitting formal analysis and clean counterexamples. Furthermore, range and long-range interactions are a topic of increasing interest for GNN researchers, with several long-range benchmarks and measurements emerging in recent years (Bamberger et al., 2025; Liang et al., 2025; Miglior et al., 2025; Zhou et al., 2025; Attali et al., 2026).

We show that existing domain-agnostic GFMs are provably bounded in range by their fixed operator bases or fixed message-passing depth, and therefore may under-reach on tasks that exceed these bounds (Section 3). We demonstrate this failure mode empirically using a simple synthetic node classification task with controllable range, for which existing methods collapse even under generous operator expansions (Section 4). Crucially, we show that this failure cannot be resolved by additive architectural changes—such as additional layers or expanded operator bases—without degrading performance on standard benchmarks. This highlights a fundamental tension: *there is no single GNN architecture that performs well across tasks with heterogeneous architectural requirements*.

Motivated by this observation, we introduce **G**raph **O**perator **B**asis **L**earning and **IN**ference (**GOBLIN**), a GFM framework for inference-time architecture adaptation (Section 5). Rather than fixing a message-passing architecture in advance, GOBLIN discovers a task-appropriate basis of linear graph operators at inference time, using only the labeled subset of nodes in the target graph (the same node label 'inpainting' task of Zhao et al. (2024) and Finkelshtein et al. (2025)). These operators are then combined via a permutation-invariant MoE, enabling zero-shot adaptation to task-specific architectural requirements without retraining or backpropagation on the target graph. While our analysis and experiments focus on range as a diagnostic axis, the proposed framework is not range-specific: the same inference-time basis learning mechanism applies to any parameterized family of graph operators encoding architectural choices. We validate the zero-shot generalization capabilities of GOBLIN on arbitrary-range synthetic tasks, as well as on a suite of real-world benchmarks, including long-range tasks (Liang et al., 2025), and demonstrate improved performance over existing domain-agnostic GFMs (Sections 4 & 6) and standard MPNNs trained end-to-end.

## 2 PROBLEM FORMULATION AND BACKGROUND

**Problem setting.** We consider *zero-shot, transductive node classification* on a single graph, following the setting of existing domain-agnostic GFMs (Zhao et al., 2024; Finkelshtein et al., 2025). Let $G = (V, E)$ be a graph with $|V| = N$ nodes, adjacency matrix $A \in \mathbb{R}^{N \times N}$, and node features $X \in \mathbb{R}^{N \times d}$. Each node $u \in V$ is associated with a (possibly unknown) label $Y_u \in \{1, \ldots, C\}$. A subset of nodes $V_L \subset V$ is labeled, with corresponding labels $Y_L \in \mathbb{R}^{N_L \times C}$ (represented as one-hot vectors), while the remaining nodes $V_U = V \setminus V_L$ are unlabeled. The task is to predict labels for all nodes in $V_U$ given $(A, X, Y_L)$, without retraining or fine-tuning on the target graph.

### 2.1 GRAPHANY

GraphAny can be interpreted as a per-node mixture of experts selector with a fixed operator basis. With GOBLIN, we will extend this pipeline to include adaptive operator selection.

Let us define a linear GNN, $\hat{Y} = SXW$, with an analytic solution given by $W^* = (SX)_L^+ Y_L$ (where $^+$ denotes the pseudo-inverse). We can think of $\hat{Y}$ as a cheap (i.e. training-free) GNN 'expert', defined entirely by the choice of graph operator $S \in \mathbb{R}^{N \times N}$, where $S$ likely encodes some function of the adjacency $A$ and likely represents some form of message passing. For example, $S = I$ is a linear layer with no mixing, and $S = A^k$ is a $k$-layer linear GNN.

By choosing a fixed operator basis $\hat{\boldsymbol{Y}} = \{\hat{Y}^{(i)}\}_{i=1}^t$, we can define a GraphAny-style per-node MoE as a summed attention weighting $\bar{Y}_u = \sum_{i=1}^t \alpha_u^{(i)} \hat{Y}_u^{(i)}$ over a given basis, where $\alpha_u = [\alpha_u^{(1)}, \alpha_u^{(2)}, \ldots, \alpha_u^{(t)}] = f_\theta(\boldsymbol{P}_u)$, and $\boldsymbol{P}_u$ denotes node-level distance features derived from $\hat{\boldsymbol{Y}}$. To ensure feature-permutation invariance, GraphAny uses node-wise linear GNN difference functions $||\hat{Y}_u^{(i)} - \hat{Y}_u^{(j)}||^2$, i.e., $t(t-1)$ features in total.[2] $f_\theta$ is an MLP that outputs per-expert logits and is trained on one or more graphs to learn to weight experts from similarity features.

Once trained, this model can be applied at inference time, zero-shot, to new graphs of arbitrary size and feature/class dimension, with no retraining and only $t$ analytic linear GNN solves. However, a trained GraphAny model is *locked to the operator basis it was trained on*, and therefore must share any shortcomings imposed by that basis. GraphAny uses $\mathcal{S} = \{S^{(1)}, \ldots S^{(t)}\} = \{I, A, A^2, (I - A), (I - A)^2\}$ (degree-normalized adjacency), i.e., a linear term plus two 'low-pass' and two 'high-pass' terms. The effective architecture, therefore, is a mixture of 0-, 1- and 2-hop message passing.

### 2.2 TRIPLE-SYMMETRY GNNS

TS-GNNs extend the GraphAny framework by formalizing GFM symmetry requirements. Specifically, TS-GNNs enforce *node equivariance*, *label equivariance*, and *feature invariance*, and implement these constraints using DeepSet constructions over nodes, features, and labels. In contrast to GraphAny's explicit MoE formulation, TS-GNNs are trained end-to-end with gradient descent.

Despite these architectural differences, TS-GNNs share a key limitation with GraphAny: they rely on a *fixed message-passing backbone*. In practice, all reported TS-GNN results use shallow architectures with at most two local message-passing layers. As a result, TS-GNNs are subject to the same under-reaching constraints as other fixed-depth GNNs: information cannot propagate beyond the receptive field determined by the chosen depth and aggregation operators.

In the following sections, we show that architectural rigidity can be explicitly demonstrated through the lens of task range, and that overcoming it requires inference-time architectural adaptation.

### 2.3 LIMITATIONS OF FIXED-BACKBONE GFMS

Both GraphAny and TS-GNN are based on message-passing architectures with a fixed depth of, at most, two layers, and must collapse via under-reaching when evaluated on tasks requiring anything beyond this. One could argue that there are trivial solutions to this: use a deep GNN or graph Transformer (GT; Dwivedi & Bresson (2020)) backbone for TS-GNN, or use an alternative or more

---

[2]Zhao et al. (2024) entropy-normalize features; for simplicity, we use squared distance features.

comprehensive basis for GraphAny. But this highlights the crucial shortcoming of both methods: any alternative architecture is still *fixed*, and there is no 'one-size-fits-all' architecture. As we see in Section 4, expanding the basis of GraphAny can help for some tasks, but introduces noise and hurts performance for others. Similarly, using a deeper GNN, or a GT, often introduces needless computational complexity for many tasks (Tönshoff et al., 2023), and can force performance trade-off against different types of tasks, often due to over-smoothing/-squashing and vanishing gradient effects (Oono & Suzuki, 2019; Alon & Yahav, 2020; Di Giovanni et al., 2023; Arroyo et al., 2025).

Put simply, a true GFM must be robust across tasks that necessitate architectural differences. Indeed, the entire motivation for developing GFMs is to avoid expensive and lengthy hyperparameter and architecture searches.[3] Existing methods do not address this; *they can only generalize to tasks that benefit from a narrow range of GNN architecture parameterizations.* Our method addresses it via adaptive basis (and therefore adaptive effective architecture) selection at inference time.

## 3 RANGE AS A DIAGNOSTIC FOR ARCHITECTURAL LIMITATIONS

In this section we formally demonstrate the rigidity of fixed-architecture GFMs, using range as a proxy. We focus on GraphAny, showing that the overall range of GraphAny can be studied by deriving the ranges of basis operators. We use the node-level range measure $\rho_u$ from Bamberger et al. (2025): given some distance metric on a graph, such as geodesic distance, range, informally, is distance-weighted pairwise node sensitivity, averaged per node. Formally,

$$\rho_u = \frac{\sum_{v \in V} J_{uv} \cdot d_G(u,v)}{\sum_{v \in V} J_{uv}} \quad \text{where } J_{uv} = \left\| \frac{\partial \phi(X)_u}{\partial X_v} \right\|_1, \tag{1}$$

and $\phi$ is some operator mapping node input features to output features, such as a GNN.

**Range of GraphAny.** As GraphAny is a weighting of linear GNNs, we can study its overall range by computing the range of individual linear GNN experts and comparing their relative weights. The Jacobian of a linear GNN decomposes as $\frac{\partial \hat{Y}}{\partial X} = \frac{\partial SX}{\partial X} W + SX \frac{\partial W}{\partial X}$. The second term requires computational means to obtain due to the pseudo-inverse, and only encodes label-fitting, not node feature mixing; we therefore focus on only the first term. To put it another way, we assume that the weights of a solved linear GNN are fixed, so $\frac{\partial W}{\partial X} = 0$ (see Appendix E for a detailed discussion of this decision). Given this, we can derive the sensitivity of an output node logit $(u,c)$ to an input node feature $(v,d')$: $\frac{\partial \hat{Y}_{uc}}{\partial X_{vd'}} = \frac{\partial}{\partial X_{vd'}} \left( \sum_{w=1}^N \sum_{k=1}^d S_{uw} X_{wk} W_{kc} \right) = S_{uv} W_{d'c}$. With reference to Eq. 1, here $J_{uv} = |S_{uv} W_{d'c}|$, and we can thus obtain the node-level range:

$$\tilde{\rho}_u = \left( \sum_{d'=1}^d \sum_{c=1}^C |W_{d'c}| \right) \left( \sum_{v \in V} |S_{uv}| \, d_G(u,v) \right) = \|W\|_1 \cdot \sum_{v \in V} |S_{uv}| \, d_G(u,v),$$

$$\text{normalizing to} \quad \rho_u = \frac{\sum_{v \in V} |S_{uv}| \, d_G(u,v)}{\sum_{v \in V} |S_{uv}|}. \tag{2}$$

Therefore, the range of a linear GNN depends only on the effective architecture, $S$, and the graph topology $A$ (via $d_G$, and possibly $S$), *not* on model weights.

GraphAny uses the degree-normalized adjacency for its operators, and as this $A$ has rows which sum to 1 (2, for the Laplacian), we obtain interpretable ranges/range upper bounds for given operators: $S = I \rightarrow \rho_u = 0$, $S = A \rightarrow \rho_u = 1$; $S = A_k$ where $(A_k)_{ij} = 1[d_G(i,j) = k] \rightarrow \rho_u = k$; $S = (I - A) \rightarrow \rho_u = 0.5$, and $S = A^k \rightarrow \rho_u \leq k$. To summarize, of the five operators used by GraphAny, three of them have range that is invariant to topology $(0, 1, 0.5)$ and the other two are upper bounded by 2 hops. This illustrates a key shortcoming in generalization. GraphAny with a basis comprised of $S = A^k$ operators can have a maximum range of $k$, and in practice, this is likely to be much lower, especially given that attention weight is shared between experts with ranges guaranteed to be $\leq 1$. With its standard basis, GraphAny *cannot solve tasks with range > 2.*

---

[3]Neural architecture search (Nunes & Pappa, 2020; Wang et al., 2023b; Dong et al., 2025) is another research area of note that is concerned with addressing this problem.

**TS-GNNs.** We cannot derive range analytically here, but the same conclusion holds; assuming an $\ell$-layer message passing backbone, $\frac{\partial \hat{Y}_{uc}}{\partial X_{vd'}} = 0$ for all $\{u, v \in V : d_G(u, v) > \ell\}$, therefore $\rho_u \leq \ell$.

**Motivation for using range as a diagnostic.** While range is not the only task property that requires architectural adaptation, it is a useful and elegant proxy here. Unlike task properties such as label homophily, range has an intuitive meaning as a property of both tasks and architectures (and, indeed, architectures *applied to* tasks), depending on whether $\phi$ is a task operator or a model. This measurable relationship between task and architectural properties makes range, as well as *known-range synthetic tasks*, useful diagnostic tools, complementary to merely evaluating downstream performance, which is predominant in the literature.

Another benefit is that GFMs that are mixtures of linear GNN experts can be decomposed into components with specific ranges using Eq. 2. As linear GNNs are entirely defined by their operator $S$, and the range of each linear GNN on a task is entirely defined by its operator and the task graph topology, range provides a geometric interpretation with which to investigate an extended, or adaptive, operator basis; for example, operators can be low- or high-pass, but they can also be short-, medium- or long-*range*.

## 4    $k$HOPSIGN: FAILURE OF A FIXED BASIS ON A KNOWN-RANGE TASK

In this section we demonstrate empirically that a fixed operator basis cannot arbitrarily adapt to tasks that require a certain minimum architecture, even when expanding the GraphAny basis.

Synthetic tasks with fixed ranges, such as RingTransfer (Bodnar et al., 2021) and $k$-Dirac (Bamberger et al., 2025) have been used in prior works to study range, but these are typically in the multi-graph setting. We adapt the core idea—nodes retrieving information at a $k$-hop distance—to the transductive setting, with $k$HopSign. Given some graph topology with diameter $> k_{\max}$ (we use a random geometric graph with 1000 nodes and radius 0.1), let each node have a scalar feature $x_u$ drawn from a standard Gaussian and binary labels given by $y_u^{(k)} = \text{sign}\left(\sum_{v \in V} \exp\left(-\frac{(d_G(u,v)-k)^2}{2\sigma_{\text{noise}}^2}\right) x_v\right)$; i.e., a 'soft' version of a fixed $k$-hop retrieval task, which collapses to $y_u^{(k)} = \text{sign}\left(\sum_{v \in \mathcal{N}_k(u)} x_v\right)$ when $\sigma_{\text{noise}} = 0$. This task has a range of $\approx k$ (exactly $k$ in the hard case). We randomly sample a 50:50 train-test split, train (and evaluate) GraphAny on Cora, and show results on $k$HopSign for $k = 1, ...8$ in Figure 1.

**Additional operator bases.** We know that GraphAny with standard operators must under-reach for $k > 2$, so to demonstrate that simply *expanding* a fixed basis does not solve the problem, we include several additional bases: further powers of the adjacency ($A^3, A^4$), precise-hop adjacencies $A_k$ (near-oracle operators for given $k$) for $k = 1, ..., 4$, and two operator families that attempt to compress varying ranges into a handful of operators; HopBins and HeatKernel. The operators for HopBins are given by $(A_{3:d^*})_{ij} = \mathbb{1}[3 \leq d(i, j) \leq d^*]$ where $d^*$ is the median geodesic distance between all node pairs; this is intended to reflect a 'binned' precise-hop basis that captures across different ranges by including (in addition to short-range 1- and 2-hop operators) a 'medium-range' term that covers the nearest ~50% of node pairs, and an $A_{>d^*}$ term that covers the remaining 'long-range' pairs. HeatKernel is intended to capture notions of short-, medium- and long-range *resistance* distance, by using the set of heat kernel terms $\{\exp(-\tau L)\}_\tau^{\text{S,M,L}}$, for temperatures $\sqrt{\tau_{\text{S}}} = 1, \sqrt{\tau_{\text{M}}} = \bar{d}, \sqrt{\tau_{\text{L}}} = 2\bar{d}$ where $\bar{d}$ is the mean geodesic node pair distance.

**Discussion.** As standard GraphAny has $\rho < 2$, it is unsurprising that it collapses to close to random guessing for $k > 1$. Even the basis including third and fourth adjacency powers collapses beyond $k > 1$, suggesting that pure smoothing is insufficient for such a task, even if its range is within the upper bound of the basis. As expected, precise-hop operators $A_k$ yield stronger performance for tasks with corresponding ranges, but this also collapses to random guessing beyond the basis (i.e. $k > 4$). Our 'binned' multi-range bases also have limited success, occasionally performing better at higher $k$, but inconsistently, and no GraphAny basis surpasses ~80% accuracy. Furthermore, if we consider Table 1, we see that when we expand the basis, though performance on $k$HopSign can improve slightly, performance *drops* significantly on the benchmarks. This demonstrates that

Table 1: Average accuracy $\Delta\%$ on 25 benchmarks for varying basis vs. standard GraphAny.

| Avg. Benchmark Acc. ($\Delta$ %) | |
| --- | --- |
| $A, A^2$ (GraphAny) | 0.0 |
| $A, A^2, A^3, A^4$ | $-5.17$ |
| $A, A_2, A_3, A_4$ | $-4.95$ |
| HopBins | $-5.29$ |
| HeatKernel | $-3.02$ |

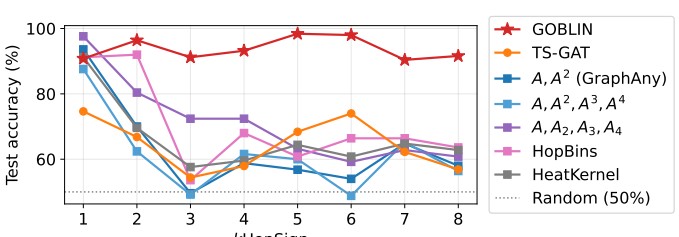

Figure 1: $k$HopSign accuracy per $k$ for varying GraphAny bases.

*there is no one-size-fits-all architecture*; widening the basis to address one architectural requirement introduces performance-hurting noise elsewhere. We require an adaptive basis. In Section 5, we introduce a novel framework, GOBLIN, that enables such a basis, and from Figure 1 we can see that it performs well on all $k$HopSign tasks, with >90% accuracy everywhere.

## 5 METHOD: GRAPH OPERATOR BASIS LEARNING & INFERENCE

We introduce GOBLIN, a framework for inference-time architecture adaptation in GFMs. GOBLIN decomposes zero-shot inference into two stages: (i) discovery of a task-adapted basis of linear graph operators from a parameterized operator family, and (ii) permutation-invariant MoE weighting over the resulting operators at the node level. Crucially, both stages operate entirely at inference time on the target graph, without retraining or backpropagation.

### 5.1 PARAMETERIZED OPERATOR FAMILIES

GOBLIN assumes access to a parameterized family of linear graph operators $S(\theta) \in \mathbb{R}^{N \times N}$, where the parameters $\theta$ control architectural properties such as receptive field, smoothing, and aggregation scale. In this work, we consider a family composed of two complementary operator classes:

$$S(\theta) = \lambda\, S^{\text{Gauss}}(\mu, \sigma) + (1 - \lambda)\, S^{\text{Heat}}(\tau), \quad \text{where} \tag{3}$$

$$S_{uv}^{\text{Gauss}}(\mu, \sigma) = \exp\left(-\frac{(\mu - d_G(u, v))^2}{2\sigma^2}\right) \quad \text{and} \quad S^{\text{Heat}}(\tau) = \exp(-\tau L_{\text{sym}}), \tag{4}$$

where $\theta = (\lambda, \mu, \sigma, \tau)$, $\lambda \in \{0, 1\}$, and $\mu, \sigma, \tau \in \mathbb{R}_+$. This approach bears similarity to existing, non-GFM works that use adaptive operators over graph scales (Frasca et al., 2020; Chien et al., 2020), including spectral graph wavelets (Hammond et al., 2011; Opolka et al., 2022). $S^{\text{Gauss}}$ corresponds to (soft) distance-localized aggregation around a target hop distance $\mu$, collapsing to exact $A_k$ when $\sigma = 0$, and $S^{\text{Heat}}$ corresponds to diffusion-based smoothing controlled by $\tau$, capturing multi-layer message passing with increasing effective range. Such an operator family allows us to span a broad spectrum of architectural behaviors, including low-, high-, and band-pass filtering, short- and long-range aggregation, and local, multi-hop (Gutteridge et al., 2023) and multi-layer message passing, with two modes, each of which encodes range via a particular distance metric: geodesic and resistance. While we focus on this family for concreteness, GOBLIN is agnostic to the specific choice of operator family, and applies to any parameterization that encodes architectural variation.

### 5.2 INFERENCE-TIME OPERATOR BASIS LEARNING

Given an operator family $S(\theta)$, the goal of the first stage is to identify a small set of operator parameters $\{\theta_i\}_{i=1}^t$ that are well-suited to the task on the target graph. For each candidate $\theta$, we define a linear GNN expert $\hat{Y}^\theta = S(\theta) X W^\theta$, $W^\theta = (S(\theta) X)_L^+ Y_L$, obtained via an analytic least-squares solution on a subset of labeled nodes, as in Zhao et al. (2024).

To evaluate operator suitability without overfitting, we partition the labeled nodes as $Y_L = Y_{L_{\text{fit}}} \cup Y_{L_{\text{eval}}}$, and define a scalar score $f(\theta) = \ell\left(\hat{Y}_{L_{\text{eval}}}^\theta, Y_{L_{\text{eval}}}\right)$, where $\hat{Y}_{L_{\text{eval}}}^\theta$ is a linear GNN fit to $Y_{L_{\text{fit}}}$ and

evaluated on $Y_{L_{\text{eval}}}$, and $\ell(\cdot)$ denotes some score metric. The function $f(\theta)$ is treated as a black-box objective over operator parameters. In practice, $f(\theta)$ may be optimized or approximated using any suitable strategy for sample-efficient black-box optimization. In our experiments, we employ Bayesian optimization (BO), using Upper Confidence Bound (UCB) sampling to trade-off exploration and exploitation. For our score metric $\ell(\cdot)$ we use 'trimmed' accuracy; mean accuracy on the set of $L_{\text{eval}}$ nodes, without the top and bottom % by margin. Design decisions are discussed further in Appendix A, but these specific choices are not essential to the framework.

Once the search budget is expended, we obtain our small operator basis $\{\theta_i\}_{i=1}^{t}$ using a greedy selector that maximizes score plus a diversity penalty (Eq. 7) to ensure we do not have redundant (i.e. effectively identical) operators.

### 5.3    PERMUTATION-INVARIANT MIXTURE OF EXPERTS

The second stage combines the adaptive operator basis via a per-node MoE. Unlike GraphAny, where the operator basis is fixed and ordered, GOBLIN must remain invariant to both operator ordering and basis size. We therefore employ a DeepSet (Zaheer et al., 2017) over operator-wise features. Like GraphAny, we employ linear-GNN-derived distance features; as we require $t$ set elements rather than a $t(t-1)$ feature vector,[4] we define per-node-pair 'disagreement' as $D_u^{(i,j)} = \|\hat{Y}_u^{(i)} - \hat{Y}_u^{(j)}\|_2^2$, and generate features that are summary statistics of this: $H_u^{(i)} = \left[ \frac{1}{t-1} \sum_{j \neq i} D_u^{(i,j)}, \quad \text{Var}_{j \neq i}\left[D_u^{(i,j)}\right], \quad \min_{j \neq i} D_u^{(i,j)}, \quad \max_{j \neq i} D_u^{(i,j)} \right]$. These features capture agreement and diversity among experts without dependence on basis size or ordering.

Note that these are simple, sensible preliminary features to illustrate the method; further experimentation would likely suggest richer expanded features. For example, we also experiment with added score features; see Appendix A.3 for details.

Next, each $H_u^{(i)}$ is embedded via a learned function (an MLP) $\phi$, producing operator-wise embeddings $E_u^{(i)} = \phi(H_u^{(i)})$. A second learned function $\psi$ maps the concatenation of $E_u^{(i)}$ and the pooled embedding $\sum_j E_u^{(j)}$ to a scalar logit $\tilde{\alpha}_u^{(i)}$, yielding normalized weights $\alpha_u^{(i)} = \exp(\tilde{\alpha}_u^{(i)}) / \sum_{j=1}^{t} \exp(\tilde{\alpha}_u^{(j)})$ for a final summation over operators. To avoid information leakage during training of $\phi$ and $\psi$, features are computed from experts fitted on $Y_{L_{\text{fit}}}$, while supervision is provided by $Y_{L_{\text{eval}}}$. At inference, experts are refit on all labels $Y_L$ before feature extraction.

## 6    EXPERIMENTS

In this section we evaluate zero-shot on real-world benchmarks, comparing GOBLIN against models trained end-to-end per task, as well as against GraphAny and TS-GNNs.

Our BO objective is a trimmed mean: accuracy on $Y_{L_{\text{eval}}}$ without the top and bottom 20% of nodes by margin, which promotes operators that perform well on a *subset* of nodes as well as overall. We run BO with a total sample budget of 25, using cross-family UCB competition between separate Gaussian processes (GPs) over $\mu$ (LinGauss) and $\sqrt{\tau}$ (LinHeat)[5], with search spaces scaled by the mean pairwise geodesic distance of the target graph. After exhausting the budget, we select a basis of $K = 4$ operators via a greedy diversity-penalized selector (Eq. 7). Full hyperparameter details and design decisions are given in Appendix A. For fair comparison with other models, we train on Cora, the same training graph used by Finkelshtein et al. (2025), and evaluate zero-shot, following the experimental procedure from that work.

We evaluate on 25 standard benchmark tasks (Table 2), as well as 4 real-world long-range tasks (Table 3) and our synthetic $k$HopSign tasks (Figure 1). The long-range tasks are from CityNetworks (Liang et al., 2025), a benchmark on the classification of road junction accessibility from large-diameter graphs ($\sim$1-600k nodes), requiring information at distances of up to 16 hops.

---

[4]In practice, we can derive features for up to `basis_size` samples to diversify the feature space and mask non-basis weights at output; see Appendix A.3.2.

[5]We search in $\sqrt{\tau}$ rather than $\tau$ as it is approximately proportional to hop distance, and spaces evaluations more densely at low $\tau$ where sensitivity is highest; see Appendix A.

Table 2: Mean $\pm$ std % accuracy (5 seeds) across 25 benchmark graph datasets. End-to-end models are trained directly on each task, so are generally easier than the zero-shot setting. Non-GOBLIN columns are reproduced from Finkelshtein et al. (2025). Heterophilic graphs are marked with an asterisk. First-/second-best results per dataset are color-coded.

| | End-to-end | | Zero-shot | | |
| --- | --- | --- | --- | --- | --- |
| | MeanGNN | GAT | GraphAny | TS-Mean | **GOBLIN** |
| Actor* | $32.03 \pm 0.29$ | $32.59 \pm 0.83$ | $27.54 \pm 0.20$ | $28.09 \pm 0.93$ | $25.36 \pm 0.14$ |
| AirBrazil | $32.31 \pm 7.50$ | $35.38 \pm 4.21$ | $33.84 \pm 15.65$ | $39.23 \pm 5.70$ | $50.77 \pm 9.55$ |
| AirEU | $39.12 \pm 6.44$ | $39.00 \pm 4.30$ | $41.25 \pm 7.25$ | $35.88 \pm 6.91$ | $41.25 \pm 3.95$ |
| AirUS | $43.93 \pm 1.16$ | $43.03 \pm 2.08$ | $43.35 \pm 1.62$ | $42.34 \pm 2.12$ | $51.78 \pm 4.43$ |
| AmzComp | $73.88 \pm 0.88$ | $70.94 \pm 3.40$ | $82.79 \pm 1.13$ | $81.37 \pm 1.25$ | $80.67 \pm 2.43$ |
| AmzPhoto | $88.95 \pm 1.08$ | $80.78 \pm 3.59$ | $89.91 \pm 0.88$ | $90.18 \pm 1.30$ | $88.90 \pm 1.34$ |
| AmzRatings* | $40.74 \pm 0.13$ | $40.63 \pm 0.66$ | $42.80 \pm 0.09$ | $42.27 \pm 1.40$ | $44.44 \pm 0.00$ |
| BlogCatalog | $84.48 \pm 0.74$ | $78.20 \pm 7.23$ | $71.54 \pm 3.04$ | $76.30 \pm 2.92$ | $75.71 \pm 2.60$ |
| Chameleon* | $61.75 \pm 0.94$ | $58.46 \pm 6.23$ | $63.64 \pm 1.48$ | $60.83 \pm 5.41$ | $67.37 \pm 0.26$ |
| Citeseer | $65.06 \pm 1.30$ | $63.92 \pm 0.84$ | $68.88 \pm 0.10$ | $68.66 \pm 0.19$ | $65.30 \pm 0.00$ |
| CoCS | $80.87 \pm 0.69$ | $82.28 \pm 0.86$ | $90.06 \pm 0.80$ | $90.92 \pm 0.47$ | $89.64 \pm 1.13$ |
| CoPhysics | $79.05 \pm 1.13$ | $85.92 \pm 1.10$ | $91.85 \pm 0.34$ | $92.61 \pm 0.61$ | $89.58 \pm 1.59$ |
| Cornell* | $63.78 \pm 1.48$ | $69.73 \pm 2.26$ | $63.24 \pm 1.32$ | $68.65 \pm 2.42$ | $67.57 \pm 0.00$ |
| DBLP | $65.01 \pm 2.40$ | $67.34 \pm 2.75$ | $71.48 \pm 1.44$ | $66.42 \pm 3.65$ | $69.68 \pm 2.67$ |
| FCora | $55.85 \pm 1.04$ | $59.95 \pm 0.88$ | $51.18 \pm 0.78$ | $53.58 \pm 0.73$ | $55.28 \pm 0.98$ |
| Minesweeper* | $84.06 \pm 0.18$ | $84.15 \pm 0.24$ | $80.46 \pm 0.11$ | $80.68 \pm 0.38$ | $81.00 \pm 0.00$ |
| Pubmed | $74.56 \pm 0.13$ | $75.12 \pm 0.89$ | $76.46 \pm 0.08$ | $74.98 \pm 0.56$ | $74.70 \pm 0.00$ |
| Questions* | $97.16 \pm 0.06$ | $97.13 \pm 0.05$ | $97.07 \pm 0.03$ | $97.02 \pm 0.01$ | $96.26 \pm 0.02$ |
| Roman* | $69.37 \pm 0.66$ | $69.80 \pm 4.18$ | $63.34 \pm 0.58$ | $66.36 \pm 1.02$ | $62.48 \pm 0.00$ |
| Squirrel* | $43.32 \pm 0.66$ | $38.16 \pm 1.04$ | $49.74 \pm 0.47$ | $41.81 \pm 0.80$ | $56.02 \pm 0.27$ |
| Texas* | $76.76 \pm 1.48$ | $81.62 \pm 6.45$ | $71.35 \pm 2.16$ | $73.51 \pm 4.01$ | $78.38 \pm 0.00$ |
| Tolokers* | $78.59 \pm 0.66$ | $78.22 \pm 0.37$ | $78.20 \pm 0.03$ | $78.12 \pm 0.09$ | $78.66 \pm 0.02$ |
| Wiki | $74.23 \pm 0.89$ | $68.91 \pm 9.50$ | $60.27 \pm 3.06$ | $69.89 \pm 1.31$ | $64.09 \pm 3.80$ |
| Wisconsin* | $74.12 \pm 12.20$ | $73.33 \pm 8.27$ | $59.61 \pm 5.77$ | $61.18 \pm 11.38$ | $82.35 \pm 0.00$ |
| WkCS | $71.97 \pm 1.70$ | $74.99 \pm 0.59$ | $74.11 \pm 0.60$ | $74.16 \pm 2.07$ | $63.42 \pm 0.42$ |
| **Average** | $66.04 \pm 1.83$ | $65.98 \pm 2.91$ | $65.76 \pm 1.96$ | $66.20 \pm 2.31$ | $\mathbf{68.03 \pm 1.42}$ |

**Discussion.** Table 2 shows that, while other methods are, on average, within 0.5% of each other, GOBLIN is about 2% better than the next best model, and beats standard MPNNs (Velickovic, 2017) trained end-to-end on the tasks. We note that GOBLIN's overall higher performance is dominated by a few datasets (AirBrazil, $>10\%$, Wisconsin, AirUS, Squirrel $>6\%$); indeed, for some tasks, GOBLIN under-performs compared to baselines. This is expected performance for a more general model: we want robust generalization across tasks, and not necessarily SOTA performance. Indeed, these results suggest that existing GFMs may not be robust to these tasks, and that they may be longer-range than expected. Figure 2, in which we report the ranges of GOBLIN on several benchmarks (selected $k$HopSign and CityNetworks, and the 5 tasks from Table 2 for which GOBLIN's accuracy improvement over all other methods is largest), validates this idea. As expected, the $k$HopSign tasks increase in range with increasing $k$. On average, the 25 standard benchmark tasks have low range, around that of a 2-hop synthetic task.[6] Our max-$\Delta\%$ tasks, however, are (with the exception of Wisconsin) much higher range than the average. Indeed, we see a weak but statistically significant relationship between performance improvement on GOBLIN and task

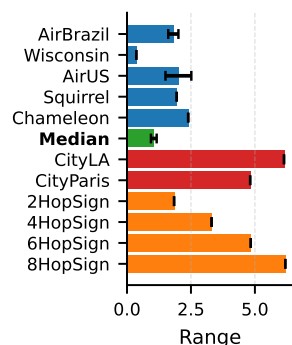

Figure 2: Range of GOBLIN on several benchmarks, $\pm\sigma$ over seeds. Median is over all 25 benchmarks.

---

[6]Median avoids significant skew due to outliers; see Appendix B for details and additional range results.

Table 3: Mean $\pm$ std % accuracy (5 seeds) on CityNetworks. We report TS-GAT as it performed better than TS-Mean.

|  | MeanGNN | GAT | GraphAny | TS-GAT | **GOBLIN** |
|---|---|---|---|---|---|
| Paris | $20.79 \pm 0.25$ | $20.73 \pm 0.28$ | $20.09 \pm 0.10$ | $16.36 \pm 1.73$ | $24.59 \pm 0.00$ |
| Shanghai | $22.86 \pm 0.13$ | $20.80 \pm 0.18$ | $19.72 \pm 0.02$ | $19.93 \pm 1.45$ | $23.53 \pm 0.00$ |
| LA | $19.34 \pm 0.08$ | $18.74 \pm 0.25$ | $17.19 \pm 0.02$ | $15.85 \pm 1.27$ | $24.29 \pm 0.00$ |
| London | $21.58 \pm 0.12$ | $21.13 \pm 0.15$ | $17.13 \pm 0.01$ | $17.60 \pm 1.49$ | $21.60 \pm 0.00$ |
| **Average** | $21.14 \pm 0.14$ | $20.35 \pm 0.21$ | $18.53 \pm 0.04$ | $17.44 \pm 1.49$ | $\mathbf{23.50 \pm 0.00}$ |

range; see Figure 5 in Appendix B. We can conclude that GOBLIN's architectural adaptivity compared to fixed-short-range baselines is the main contributor to the performance boost.[7]

**Long-range.** In Table 3 we see that GOBLIN out-performs baseline GFMs by ~3–8%, an encouraging result, and one we would expect, given their strictly low ranges. GOBLIN also out-performs end-to-end trained MPNNs overall, but we note that (for fair cross-benchmark comparison) these use the same parameter search described above, with a fixed depth of 2 layers. In practice, CityNetworks is extremely depth-sensitive, and all of these results can be significantly improved upon by increasing model depth. We discuss this in detail in Appendix C.2.

## 7 CONCLUSION

This work aims to highlight that a prerequisite for an effective GFM (fixed architecture *and* model weights) is a versatile *architecture*; for GNNs, the problem of choosing such a foundation architecture arguably remains unsolved. Range is one axis along which architecture is significant; we highlight the weakness of established GFMs in generalization along this axis due to their static architectures. We encourage the community to include synthetic and real-world long-range tasks in future evaluation of GFMs.

**Limitations.** GOBLIN is not a SOTA method; as a MoE architecture composed of weak linear GNN learners, it is insufficiently expressive for many complex tasks (see Appendix C). Our aim is to highlight that in GFM development and evaluation, architectural trade-offs should be explicitly considered, and pure-performance homogeneous-task benchmarking is insufficient.

GOBLIN incurs much higher inference-time cost than fixed-basis methods like GraphAny, due to both additional linear GNN solves and costlier operator construction—though it is still much faster than end-to-end training. See Appendix D for a detailed discussion.

**Future work.** Our current instantiation of GOBLIN uses simple, heuristically chosen sampling rules for operator selection and a small set of summary features for MoE weighting. These choices are not fundamental to the framework. More expressive operator features would likely improve performance, and adaptive sampling strategies could balance search budget against task difficulty. An investigation into training across *multiple graphs* would likely lead to improvements in both efficiency and generalization.

ACKNOWLEDGMENTS

B.G. is supported by the EPSRC Centre for Doctoral Training in AIMS (EP/S024050/1). M.B. is supported by the EPSRC Turing AI World-Leading Research Fellowship No. EP/X040062/1 and EPSRC AI Hub No. EP/Y028872/1.

---

[7]We note that strong performance on these particular tasks is *not* a simple consequence of heterophily; indeed, AirBrazil is homophilic. Contrast this with Roman, a large-diameter heterophilic graph that nevertheless has very low range; see Figure 3, Bamberger et al. (2025) and Arnaiz-Rodriguez & Errica (2025).

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

# A  GOBLIN PIPELINE: ADDITIONAL DETAILS

This section contains more detailed explanations of the operator search and DeepSet training stages of the GOBLIN pipeline, as well as explaining our GOBLIN hyperparameter search space and design decisions.

## A.1  OPERATOR SEARCH

The detailed steps for the operator search are as follows:

1. If $\mu$, $\sqrt{\tau}$ scaling factors are non-zero, compute their maximum values by multiplying by the dataset's mean all-pairs shortest-path distance. (As we already have APSPDs cached for generating LinGauss operators, this is free.) Then seed each family's search space with a set of anchor samples spread across $[0, \mu_{\max}]$ or $[0, \sqrt{\tau}_{\max}]$ respectively.

2. Until the sample budget is expended, select the next operator to evaluate via cross-family UCB competition. A separate GP is maintained for each family, one fitted to observed $(\mu, \text{accuracy})$ pairs for LinGauss, one to $(\sqrt{\tau}, \text{accuracy})$ pairs for LinHeat. At each step, each GP proposes its locally-optimal parameter:

$$\mu^* = \underset{\mu \, \in \, [0, \mu_{\max}] \setminus \mathcal{O}_\mu}{\arg\max} \Big[ m_\mu(\mu) + \beta \, s_\mu(\mu) \Big], \tag{5}$$

$$\sqrt{\tau}^* = \underset{\sqrt{\tau} \, \in \, [0, \sqrt{\tau}_{\max}] \setminus \mathcal{O}_\tau}{\arg\max} \Big[ m_\tau(\sqrt{\tau}) + \beta \, s_\tau(\sqrt{\tau}) \Big], \tag{6}$$

where $m_\mu$, $s_\mu$ and $m_\tau$, $s_\tau$ are the posterior mean and standard deviation of the LinGauss and Heat GPs respectively, $\mathcal{O}_\mu$ and $\mathcal{O}_\tau$ are the sets of already-evaluated parameters for each family (masked to $-\infty$), and $\beta \geq 0$ controls the exploration–exploitation trade-off. Whichever proposal attains the higher acquisition value wins the sample and is evaluated. This cross-family competition directs the shared budget toward whichever family and parameter region currently appears most promising.

3. Select a basis of size $K = \texttt{basis\_size}$ from the full set of sampled operators $\mathcal{O}_\mu \cup \mathcal{O}_\tau$ by greedily repeating:

$$\theta^* = \underset{\theta \, \in \, (\mathcal{O}_\mu \cup \mathcal{O}_\tau) \setminus \Theta}{\arg\max} \Big[ \ell(\theta) - \lambda \cdot \max_{\theta' \in \Theta} \cos\left( \hat{Y}^\theta, \hat{Y}^{\theta'} \right) \Big], \tag{7}$$

and adding $\theta^*$ to $\Theta$, until $|\Theta| = K$. Here $\ell(\theta)$ is the BO score for the operator parameterized by $\theta$, $\hat{Y}^\theta$ is its $L_2$-normalized prediction vector on the train-eval split, and $\lambda \geq 0$ is the diversity penalty weight. Setting $\lambda = 0$ recovers pure top-$K$ selection.

## A.2  TRAINING

During development/during our hyperparameter search we explored two training modes: `stochastic` and `pool`. `stochastic` is essentially the same as the training mode used by GraphAny; using a training split of a single graph, each epoch samples over a random minibatch of the labeled training nodes and splits them into 'target' and 'reference' nodes.

For a single seed, a GOBLIN DeepSet component trained in this way will be trained only on a single set of basis features; the features derived from the basis obtained using the same sampling strategy applied at inference time. This is how GraphAny works; they train an MLP on a consistent set of features derived from the same 5 linear operators. Because our model supports arbitrary graph operators, we also experiment with `pool` training. In this setup, for our training graph (Cora, in our case) we sample a grid of linear operators across the $\mu$, $\sqrt{\tau}$ feature space; 25 of each, 50 operators total. Then during each epoch, rather than using the best-performing operators (as we do at inference time), we sample a random set of operators from this pool. This ensures that the DeepSet learns from a diverse array of operator-derived features. Our hyperparameter search found that `pool`-based training was more effective than `stochastic`.

### A.3 Hyperparameter glossary

This section contains explanations for terms used in the hyperparameter search table in Appendix A.4.

#### A.3.1 Operator search and GP

**BO objective: `trimmed_20`.** Per-node accuracy on $Y_{L_{\text{eval}}}$, sorted by margin, with the top and bottom 10/20% removed. Accuracy is also standardized so random guessing is at zero, perfect accuracy is at 1, and worse than random guessing is negative. We use trimmed accuracy to promote the sampling of operators that perform well on substantial subsets of nodes, rather than exclusively trying to find only the best 'all-rounder' operators.

**$\mu, \sqrt{\tau}$ scale factors.** Option to determine operator search space at inference time based on graph properties. Default (scale factor $= 0$) is to use a fixed scale factor for all datasets of $[0, 8.0]$ and $[0, 5.0]$ for $\mu, \sqrt{\tau}$ respectively. For $> 0$ the $\mu, \sqrt{\tau}$ max at inference time is computed by average pairwise shortest-path distance multiplied by a hyperparameter scale factor. We search in $\sqrt{\tau}$ rather than $\tau$ because (i) $\sqrt{\tau}$ is approximately proportional to hop distance (as diffusion variance scales linearly with $\tau$), and (ii) because the low-$\tau$ regime is both more practically important and more sensitive to parameter values, thus making uniform spacing in $\sqrt{\tau}$ preferable.

**$\mu, \sqrt{\tau}$ anchors.** In order to prevent UCB sampling from falling into local maxima, we seed the parameter space with diverse, reasonable initial samples. In practice, $\sqrt{\tau}$ is a narrower space, so we use fewer anchors, and $\mu$ is a wider space, with terms decaying fairly quickly far from $\mu$, so we use more. The value given is the number of anchors sampled uniformly between $\mu_{\min}, \sqrt{\tau}_{\min}$ (both zero in practice) and $\mu_{\max}, \sqrt{\tau}_{\max}$.

#### A.3.2 DeepSet

**Training mode.** See Appendix A.2

> **`stochastic`.** Fixed-operator minibatch-style training, similar to GraphAny's
>
> **`pool`.** Train the DeepSet with randomly sampled operators per epoch for feature diversity

**Weight selection** Which operators the DeepSet sees as input, and how the output weights are post-processed. For `pre_filter` and `mask_by_deepset` variants, a suffix of `_half` or `_all` determines the feature set size: whether to use only the top 50% of operators (by BO score) or the full non-redundant set as DeepSet input.

> **`standard`.** DeepSet input is derived from basis size (in practice, 4) operators *only*; $\alpha$ weights derived from these as explained in Section 5.3.
>
> **`pre_filter_{half,all}`.** DeepSet sees the selected operator set at input, generates summary-statistic per-operator features over the full (non-redundant; see 'diversity penalty $\lambda$') or top-half operator set, and produces outputs for all of these operators, but final inference is only over the selected basis; non-selected operators are masked at softmax stage. This method ensures that final prediction/weighting is over a small set of high-performance operators, but the feature space is more diverse. The most effective method overall when using the full set of non-redundant operators.
>
> **`mask_by_deepset_{half,all}`.** Like `pre_filter`, the DeepSet sees the selected operator features, but rather than the output weight masking being pre-determined by BO scores, the final basis set is selected by taking the top DeepSet outputs and masking the others. Gives more responsibility to the DeepSet; it both weights basis operators, and has to identify the best ones. In practice this is less effective than `pre_filter`.

**Attention temp.** Pre-softmax weighting for $\alpha$; higher values smooth the softmax more

**Score feature.** Whether to include an additional feature along with the operator similarity statistics described in Section 5.3.

> **`none`.** Pure similarity features only, as in GraphAny.

**trimmed.** Appends the (top and bottom 20%) trimmed score that the GP fits as an additional per-operator feature, encoding not just similarities between operators but also which operators perform well on a small set of sample nodes. In practice, pure similarity features suffice.

## A.4 HYPERPARAMETER SEARCH

We performed a random hyperparameter search for GOBLIN overnight over ~800 configs, each evaluating in ~7-8 minutes, each training on Cora (seed 0) and evaluating on a subset of datasets. The search space and final hyperparameters for the best config are given in Table 4. See Appendix A.3 for term definitions.

Table 4: Hyperparameter search space for GOBLIN, used in Figure 1 and Tables 2 & 3.

| | **Hyperparameter** | **Search space** | **Chosen** |
|---|---|---|---|
| *Operator search* | Train dataset | — | Cora |
| | BO objective | — | `trimmed_20` |
| | `sample_budget` | 20, 25, 30, 35 | 25 |
| | UCB $\beta$ | 2.0, 3.0, 4.0 | 3.0 |
| | `basis_size` | 3, 4, 5 | 4 |
| | Diversity penalty $\lambda$ | 0.0, 0.1, 0.2, 0.3, 0.4, 0.5 | 0.2 |
| | $\mu$ scale factor | None (default [0., 8.]), 1.25, 1.5, 2.0 | 1.25 |
| | $\sqrt{\tau}$ scale factor | None (default [0., 5.]), 1.0, 1.25, 1.5 | 1.25 |
| | $\mu$ anchors | None, 2, 3, 5 | 5 |
| | $\sqrt{\tau}$ anchors | None, 1, 2 | 1 |
| | Additional anchors | None, $A^2$ | $A^2$ |
| *GP* | RBF length scale | — | 1.0 |
| | White noise | — | 0.2 |
| *DeepSet* | Training mode | `stochastic`, `pool` | `pool` |
| | #Batches | 500, 1000, 2000 | 500 |
| | Weight selection | `standard`, `pre_filter {_half, _all}`, `mask_by_deepset{_half, _all}` | `pre_filter_all` |
| | Score feature | `none`, `trimmed` | `none` |
| | Hidden dim. | 32, 64 | 64 |
| | Attention temp. | 1.0, 2.0, 5.0, 10.0 | 2.0 |
| | #DeepSet layers | 2, 3 | 3 |
| | #Head layers | 1, 2 | 1 |
| | Learning rate | $5 \times 10^{-5}, 10^{-4}, 3 \times 10^{-4}$ | $3 \times 10^{-4}$ |
| | Dropout | 0.0, 0.1, 0.2 | 0.1 |

## A.5 GAUSSIAN PROCESS DISCUSSION

Although the evaluation metric is discrete, the objective varies smoothly with operator parameters in practice, since predictions depend continuously on the underlying graph operators and on the corresponding least-squares solutions. Small perturbations of operator parameters typically induce only small changes in node logits, and hence only limited changes in aggregate accuracy, with abrupt changes arising only when a small number of nodes lie close to decision boundaries. To account for this behavior, we model the objective using a Gaussian process with an RBF kernel, encoding the assumption that operators with similar parameters tend to exhibit correlated performance. We include an additive white-noise term to capture residual non-smoothness and evaluation noise arising from finite label sets and discrete predictions. In our implementation we fix the kernel length scale to 1 and the noise variance to $\sigma_n = 0.2$, which we found to provide a stable inductive bias without overfitting. We further reduce sensitivity to unstable decision boundaries by using a trimmed-mean accuracy over nodes as the optimization objective, which empirically improves robustness to small, localized prediction sensitivity.

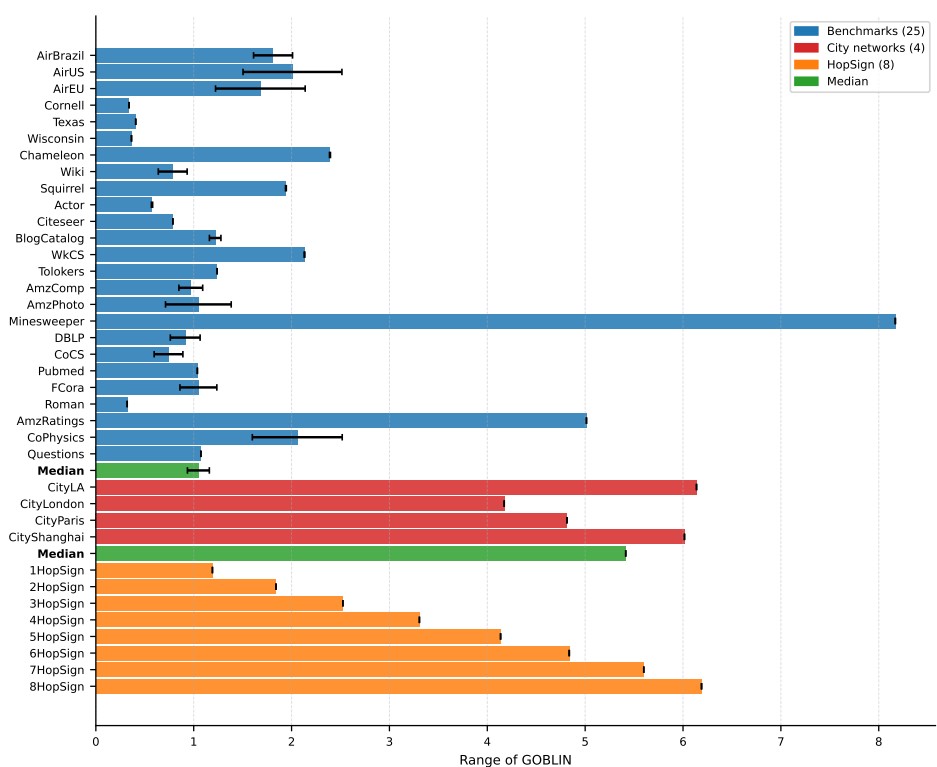

Figure 3: Range of GOBLIN on synthetic and real-world benchmark tasks ($\pm\sigma$ over 5 seeds).

## B RANGE OF GOBLIN: ADDITIONAL RESULTS AND DISCUSSION

Figure 3 shows overall GOBLIN model ranges for each real-world and synthetic task, as well as median ranges for the 25 benchmarks and CityNetwork benchmarks. This range is an aggregate over the ranges of individual operator ranges, weighted by average $\alpha$ over all nodes. Details below.

Figure 4 is similar, but rather than including a GOBLIN range over aggregated over all operator bases and all seeds, this plot shows the *best-performing single operator* per dataset across all seeds (i.e., only a single seed per dataset). This is a useful counterpoint to Figure 3 for several reasons. Firstly, it shows the ranges of operators that were actually effective; as a result we see that the range per $k$HopSign task in Figure 4 consistently tracks $k$ closely, whereas in Figure 3 it is slightly lower due to the presence of additional noisy operators that reduce the range without affecting performance. We also observe this for the CityNetworks tasks; the best-performing single operators have ranges of 12-15 hops, but GOBLIN on average is ~4-6. Secondly, looking at best-performing hops removes the confounding impact of ineffective outlier operators. For example, in Figure 3 Minesweeper is an outlier with a very large range. This occurs because Minesweeper has a large average APSPD (~50 hops; see Table 6), so our scale factor hyperparameters (see Appendix A.1) result in a large maximum $\mu$, which may contribute enormously to the overall range without necessarily improving performance. Figure 4 reveals that Minesweeper's task range is likely much lower in practice: the best-performing overall hyperparameter is LinHeat$(1.48)$.

**Computing GOBLIN range from per-node $\alpha$ and operator ranges.** Given node level weights $\boldsymbol{\alpha} \in \mathbb{R}^{N \times t}$ for our $t$-operator basis, we compute per-expert average weights $\bar{\alpha} = \frac{1}{N}\mathbf{1}^\top \boldsymbol{\alpha} \in \mathbb{R}^t$.

Similarly, we obtain graph-level per-expert ranges by averaging over node-level ranges $\boldsymbol{\rho}_u \in \mathbb{R}^t$: $\boldsymbol{\rho}_G = \frac{1}{N}\sum_{u \in V} \boldsymbol{\rho}_u$, where each element $\boldsymbol{\rho}_u$ is as defined in Equation 2.

Figure 3 plots $\bar{\alpha}^\top \boldsymbol{\rho}_G$, averaged over seeds, $\pm$ standard deviation.

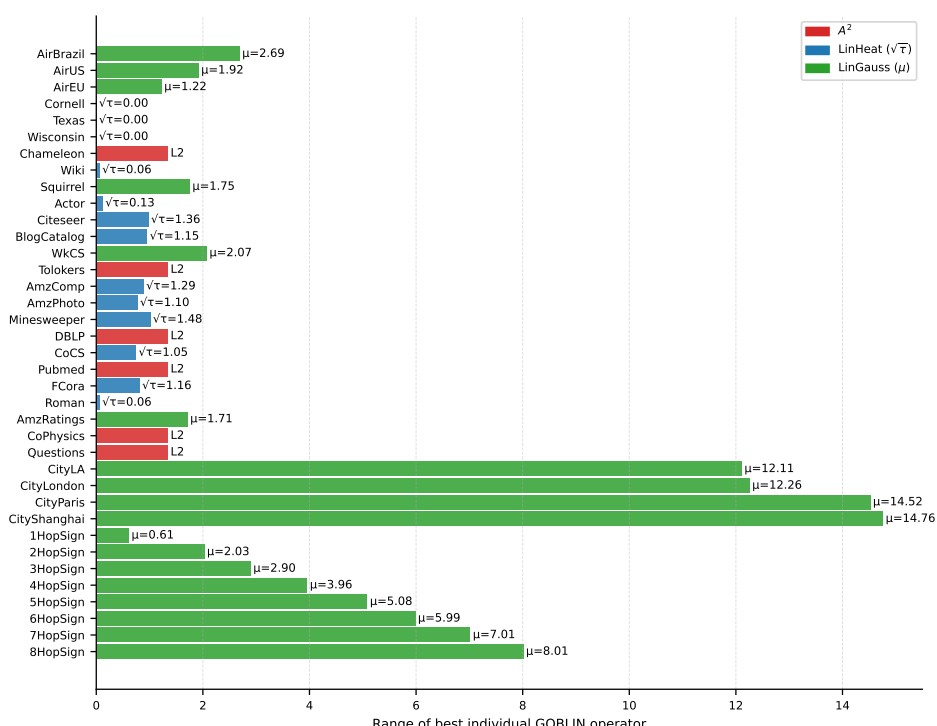

Figure 4: Range of best-performing operator on each task, out of all operators selected by GOBLIN across all 5 seeds.

**Discussion.** As expected, the range of GOBLIN on $k$HopSign tasks consistently increases with $k$. For other tasks there is significant variation depending on the task. While there are exceptions (Minesweeper and Wisconsin especially), we observe that, in general, the tasks on which GOBLIN most strongly out-performs other models tend to be more likely to have larger range. This suggests that GOBLIN is indeed learning a higher-range adaptive architecture to improve performance on tasks that are more challenging to standard shallow MPNNs and shallow MPNN-based GFMs with low ranges.

**Correlation between task range and GOBLIN vs. non-GOBLIN performance margin.** To expand on the point made above; we observe a moderate but statistically significant positive correlation (Spearman $\rho = 0.41, p = 0.025$) between a task's range using GOBLIN and GOBLIN's accuracy advantage over the strongest competing method (out of (end-to-end) MeanGNN, GAT, (zero-shot) GraphAny, TS-GNN, as in Tables 2) & 3). Datasets on which GOBLIN learns to aggregate over longer spatial scales tend to be those where it achieves the largest gains, consistent with the hypothesis that long-range structure is a factor driving GOBLIN's relative advantage. The correlation is modest rather than deterministic, reflecting that range is one of several factors (e.g. feature informativeness, and graph topology), and that the set of datasets evaluated over are mostly short-range.

### B.1 RANGE OF GOBLIN OPERATOR FAMILIES

**Gaussian.** By definition, the range of $S^{\text{Gauss}}$ operators is centered on $\mu$. Depending on topology and $\sigma$, the range may be higher or lower than $\mu$: $\rho > \mu$ when $|\mathcal{N}_{k+\delta}(u)| > |\mathcal{N}_{k-\delta}(u)|$, and vice versa; exactly $\mu$ for, for example, regular lattices.

**Heat.** The heat kernel operator $S^{\text{Heat}}(\tau)$ spreads information from a node via a diffusion process rather than targeting a specific distance. Its effective range corresponds to the typical displacement of a random walk after time $\tau$. For small $\tau$, aggregation remains localized; as $\tau$ increases, the range

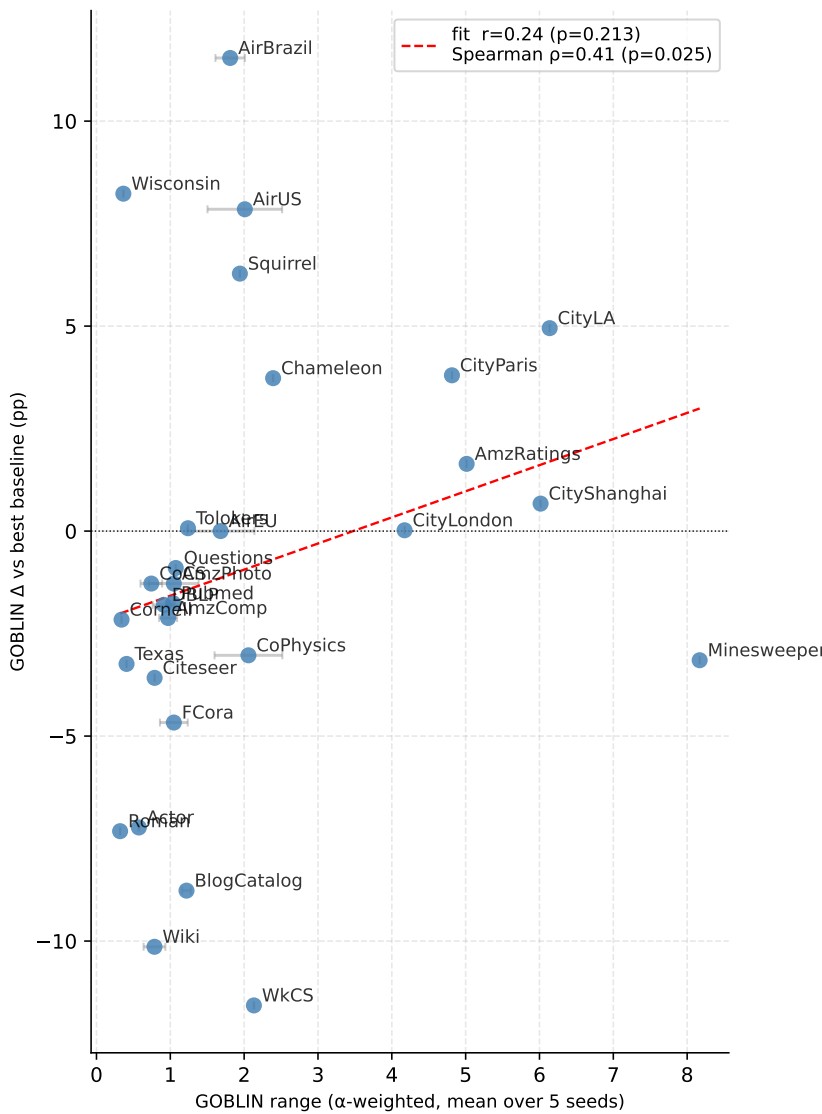

Figure 5: The relationship between task range (as captured by GOBLIN) and GOBLIN performance improvement (Δ average % accuracy) over non-GOBLIN models (MPNNs, GraphAny and TS-GNN, as in Tables 2 & 3) for 29 real-world benchmarks.

grows gradually due to the diffusive nature of random walks, and eventually saturates once the walk mixes across the graph. As a result, the heat operator provides smooth, multi-scale aggregation but does not offer direct control over a precise interaction range. To put it another way, it represents a range-driven operator using random-walk/resistance distance as the distance metric on the graph, rather than geodesic distance.

## C  DATASET DETAILS

### C.1  DATASETS

We use 30 datasets in this paper (29 evaluation benchmarks and 1 training graph, Cora): Cornell, Texas, Wisconsin, Chameleon, Squirrel, Actor (Pei et al., 2020), AirBrazil, AirUS, AirEU (Ribeiro et al., 2017), Wiki, BlogCatalog (Yang et al., 2023), Cora, Citeseer, Pubmed Yang et al. (2016), AmzPhoto, AmzComp, CoCS, CoPhysics (Shchur et al., 2018), WikiCS (Mernyei & Cangea, 2020),

FullCora, DBLP Bojchevski & Günnemann (2017), Roman, AmzRatings, Minesweeper, Tolokers and Questions (Platonov et al., 2023), and finally the four CityNetworks tasks: (City)London, Paris, LA and Shanghai (Liang et al., 2025).

With the exception of CityNetworks, we draw from the same set of benchmarks as Finkelshtein et al. (2025), but drop LastFMAsia and Deezer (Rozemberczki et al., 2021) as they can no longer be downloaded from Pytorch Geometric (Fey & Lenssen, 2019), as well as Arxiv (Hu et al., 2020), as its size makes it impractical to work with. Full dataset statistics are in Table 6.

## C.2    CITYNETWORKS DISCUSSION

See Table 3; we see that both GraphAny and TS-GNNs struggle, due to the long-range nature of the task and those models' fixed short-range architectures. While performance increases could be obtained by retraining the GFM with more layers, this (i) defeats the purpose of a GFM and (ii) would increase computational complexity (and likely decrease performance accuracy on short-range tasks). Using our GOBLIN parameterization we obtain results at least 2-3% better than other GFMs, and better than or on-par with GNNs trained explicitly on each dataset.

Table 5: CityNetworks accuracy (%) for deep ($L = 16$) MeanGNN and GAT trained directly on each dataset using the hyperparameter search described in Finkelshtein et al. (2025). Results are mean $\pm$ std over 4 training seeds with fixed data splits.

| Dataset | MeanGNN | GAT |
|---|---|---|
| Paris | $27.10 \pm 0.67$ | $46.75 \pm 0.50$ |
| Shanghai | $28.64 \pm 0.36$ | $58.50 \pm 0.75$ |
| LA | $26.89 \pm 0.15$ | $53.36 \pm 0.59$ |
| London | $28.13 \pm 0.25$ | $47.72 \pm 0.25$ |

### C.2.1    DEPTH

As shown in Liang et al. (2025), deeper models perform much better at the CityNetworks tasks. It is likely that increasing depth would improve performance for TS-GNN, and we know it increases performance significantly for non-zero-shot MPNN baselines. In Table 5 we confirm this explicitly; using identical hyperparameters to the MPNN results in Table 3, we only change the depth from $L = 2$ to $L = 16$, and observe a huge performance improvement, especially for GAT.

While GOBLIN's maximum range in our particular hyperparameterization is 8, in comparison to an 'oracle' MPNN with 16 layers, it is likely that even an expanded GOBLIN would struggle on this task. Simply put, it is too complex for weak linear GNN experts. Rather than undermining our method though, we believe this finding highlights our central argument: we are not trying to argue that GOBLIN is SOTA—indeed, other works exist which already improve on many of the benchmark results in this paper (Hayler et al., 2025). CityNetworks is highly sensitive to a particular architecture/hyperparameter; TS-GNNs and GraphAny fail to generalize to it, so in a real setting we would be forced to do an expensive, time consuming architecture search, training a new MPNN from scratch each time, to learn that $L = 16$ is optimal. GOBLIN cannot currently achieve performance on par with the best results reported by Liang et al. (2025) due to its *own* architectural limitation: expressiveness. But future GFM research can be more mindful of this tradeoff. More expressive MoE models can hypothetically generalize to different task types and architectural requirements without needing a one-size-fits-all. We leave such investigations to future work.

Table 6: Statistics for the 30 datasets used in this paper. APSPD is all-pairs shortest path (geodesic) distance.

| Dataset | #Nodes | #Edges | #Feature | #Classes | #Labeled Nodes | Diam. | Mean APSPD | Train/Val/Test Ratios (%) | Hetero-philic | Source |
|---|---|---|---|---|---|---|---|---|---|---|
| AirBrazil | 131 | 1074 | 131 | 4 | 80 | 4 | 2.17 | 61.1/19.1/19.8 | | Ribeiro et al. (2017) |
| Cornell | 183 | 554 | 1703 | 5 | 87 | 8 | 3.18 | 47.5/32.2/20.2 | ✓ | Pei et al. (2020) |
| Texas | 183 | 558 | 1703 | 5 | 87 | 8 | 3.02 | 47.5/31.7/20.2 | ✓ | Pei et al. (2020) |
| Wisconsin | 251 | 900 | 1703 | 5 | 120 | 8 | 3.25 | 47.8/31.9/20.3 | ✓ | Pei et al. (2020) |
| AirEU | 399 | 5995 | 399 | 4 | 80 | 5 | 2.32 | 20.1/39.8/40.1 | | Ribeiro et al. (2017) |
| AirUS | 1190 | 13599 | 1190 | 4 | 80 | — | — | 6.7/46.6/46.6 | | Ribeiro et al. (2017) |
| Chameleon | 2277 | 36101 | 2325 | 5 | 1092 | 11 | 3.56 | 48.0/32.0/20.0 | ✓ | Pei et al. (2020) |
| Wiki | 2405 | 17981 | 4973 | 17 | 340 | — | — | 14.1/42.9/43.0 | | Yang et al. (2023) |
| Cora | 2708 | 10556 | 1433 | 7 | 140 | — | — | 5.2/18.5/36.9 | | Yang et al. (2016) |
| Citeseer | 3327 | 9104 | 3703 | 6 | 120 | — | — | 3.6/15.0/30.1 | | Yang et al. (2016) |
| BlogCatalog | 5196 | 343486 | 8189 | 6 | 120 | 4 | 2.51 | 2.3/48.8/48.8 | | Yang et al. (2023) |
| Squirrel | 5201 | 217073 | 2089 | 5 | 2496 | 10 | 3.09 | 48.0/32.0/20.0 | ✓ | Pei et al. (2020) |
| Actor | 7600 | 30019 | 932 | 5 | 3648 | 12 | 4.11 | 48.0/32.0/20.0 | ✓ | Pei et al. (2020) |
| AmzPhoto | 7650 | 238162 | 745 | 8 | 160 | — | — | 2.1/49.0/49.0 | | Shchur et al. (2018) |
| Minesweeper | 10000 | 78804 | 7 | 2 | 5000 | 99 | 46.66 | 50.0/25.0/25.0 | ✓ | Platonov et al. (2023) |
| WikiCS | 11701 | 431206 | 300 | 10 | 580 | — | — | 5.0/15.1/49.9 | | Mernyei & Cangea (2020) |
| Tolokers | 11758 | 1038000 | 10 | 2 | 5879 | 11 | 2.79 | 50.0/25.0/25.0 | ✓ | Platonov et al. (2023) |
| AmzComp | 13752 | 491722 | 767 | 10 | 200 | — | — | 1.5/49.3/49.3 | | Shchur et al. (2018) |
| DBLP | 17716 | 105734 | 1639 | 4 | 80 | — | — | 0.5/49.8/49.8 | | Bojchevski & Günnemann (2017) |
| CoCS | 18333 | 163788 | 6805 | 15 | 300 | 24 | 5.43 | 1.6/49.2/49.2 | | Shchur et al. (2018) |
| Pubmed | 19717 | 88648 | 500 | 3 | 60 | 18 | 6.34 | 0.3/25.5/51.5 | | Yang et al. (2016) |
| FullCora | 19793 | 126842 | 8710 | 70 | 1400 | — | — | 7.1/46.5/46.5 | | Bojchevski & Günnemann (2017) |
| Roman Empire | 22662 | 65854 | 300 | 18 | 11331 | 6824 | 2331.5 | 50.0/25.0/25.0 | ✓ | Platonov et al. (2023) |
| AmzRatings | 24492 | 186100 | 300 | 5 | 12246 | 46 | 16.24 | 50.0/25.0/25.0 | ✓ | Platonov et al. (2023) |
| CoPhysics | 34493 | 495924 | 8415 | 5 | 100 | 17 | 5.16 | 0.3/49.9/49.9 | | Shchur et al. (2018) |
| Questions | 48921 | 307080 | 301 | 2 | 24460 | 16 | 4.29 | 50.0/25.0/25.0 | ✓ | Platonov et al. (2023) |
| CityParis | 114127 | 365022 | 37 | 10 | 11412 | — | — | 10.0/10.0/80.0 | | Liang et al. (2025) |
| CityShanghai | 183917 | 524184 | 37 | 10 | 18391 | — | — | 10.0/10.0/80.0 | | Liang et al. (2025) |
| CityLA | 240587 | 683046 | 37 | 10 | 24058 | — | — | 10.0/10.0/80.0 | | Liang et al. (2025) |
| CityLondon | 568795 | 1513004 | 37 | 10 | 56879 | — | — | 10.0/10.0/80.0 | | Liang et al. (2025) |

# D  TIME COMPLEXITY

We compare wall-clock time for GOBLIN against GAT baselines on the four CityNetworks tasks. All runs use 5 seeds on an A10 GPU. GOBLIN uses the hyperparameters from the random search described in Appendix A.4, trained on Cora. GAT baselines use a 4-config hyperparameter grid ($\mathrm{lr} \in \{5 \times 10^{-4}, \ 10^{-3}\} \times \mathrm{hidden\_dim} \in \{64, 128\}$, 1 seed); the best configuration for all four datasets was $\mathrm{lr} = 5 \times 10^{-4}$, $\mathrm{hidden\_dim} = 128$. GAT $L = 16$ uses the same hyperparameters as GAT $L = 2$ without a separate search. GAT models are trained independently per city per seed.

Table 7 reports per-dataset training and inference times. GOBLIN's inference cost (5–18s per city) is dominated by the Bayesian optimisation operator search rather than the DeepSet forward pass, and its training on Cora is a one-time cost shared across all downstream tasks. GAT $L = 2$ trains quickly per city but incurs repeated per-task retraining; its total wall-clock ($\sim$465s without hyperparameter search, $\sim$768s including it) is comparable to or exceeds GOBLIN's $\sim$340s despite lower accuracy on all four datasets. GAT $L = 16$ achieves substantially higher accuracy at roughly $26\times$ the wall-clock cost of GOBLIN ($\sim$8,853s, approximately 2.5 hours), consistent with $\mathcal{O}(L)$ scaling of forward and backward passes on sparse graphs.

Table 7: Wall-clock times (mean $\pm$ std over 5 seeds, A10 GPU) on CityNetworks. GOBLIN's train time (shown for LA only) is a one-time cost on Cora shared across all cities; inference includes the full BO operator search. GAT inference is a single forward pass; training is repeated per city per seed. The summary table gives total wall-clock across all 5 seeds and 4 cities, with and without GAT hyperparameter search (GOBLIN has no per-task search).

| Model | Dataset | $N$ | Hparam search | Train / seed | Inference / seed |
|---|---|---|---|---|---|
| GOBLIN (zero-shot) | LA | 240,587 | — | $19.9 \pm 0.3$s | $15.8 \pm 0.5$s |
| | London | 568,795 | — | — | $10.2 \pm 0.2$s |
| | Paris | 114,127 | — | — | $4.5 \pm 0.1$s |
| | Shanghai | 183,917 | — | — | $18.0 \pm 0.5$s |
| GAT $L = 2$ | LA | 240,587 | 67.1s | $20.5 \pm 0.2$s | $15.1 \pm 0.1$ms |
| | London | 568,795 | 149.9s | $46.1 \pm 0.4$s | $34.9 \pm 0.2$ms |
| | Paris | 114,127 | 36.6s | $11.1 \pm 0.1$s | $8.0 \pm 0.0$ms |
| | Shanghai | 183,917 | 50.0s | $15.4 \pm 0.1$s | $11.4 \pm 0.1$ms |
| GAT $L = 16$ | LA | 240,587 | | $344.7 \pm 1.2$s | $195.4 \pm 0.8$ms |
| | London | 568,795 | Non-$L$ hparams same as $L = 2$ | $792.1 \pm 2.8$s | $449.9 \pm 1.6$ms |
| | Paris | 114,127 | | $175.7 \pm 0.6$s | $97.8 \pm 0.4$ms |
| | Shanghai | 183,917 | | $258.3 \pm 0.9$s | $144.7 \pm 0.5$ms |

| | Total (5 seeds, 4 tasks) | |
|---|---|---|
| Model | Excl. search | Incl. search |
| GOBLIN | $\sim$340s | $\sim$340s |
| GAT $L = 2$ | $\sim$465s ($1.4\times$) | $\sim$768s ($2.3\times$) |
| GAT $L = 16$ | $\sim$8,853s ($26\times$) | $\sim$9,156s ($27\times$) |

**GOBLIN vs. GraphAny.**  As noted in Section 7, GOBLIN incurs higher inference-time cost than GraphAny. This overhead comes from two sources.

**More linear GNN solves.**  Time complexity for both models is dominated by linear GNN solves, and GOBLIN requires several times more solves — for our best-performing config, 25.

**Costlier operator construction.**  GraphAny operators are simple adjacency powers, which are cheap to compute. GOBLIN's $S^{\mathrm{Gauss}}$ operators require computation of all-pairs shortest-path distances (APSPD) up to an effective radius of approximately $\mu + 3\sigma$. Computing the full APSPD matrix is time- and memory-intensive, but it need only be done once per graph and can then be cached and reused across all subsequent inference calls. Furthermore, memory and time complexity on large graphs can be substantially reduced by truncating hop distances at a reasonable threshold;

in practice this makes a significant difference — very large graphs such as CityNetworks process in minutes rather than hours. $S^{\text{Heat}}$ operators are also more costly than simple adjacency powers, though the heat kernel can be efficiently approximated using a truncated Taylor expansion of the matrix exponential, significantly reducing this overhead.

These limitations should not be underestimated. However, it is crucial to remember that (i) GOBLIN is still significantly more computationally efficient than end-to-end model training, and (ii) in practice, GOBLIN's search complexity can be reduced; it is high in this paper to ensure robustness across different tasks. GOBLIN could be reduced to GraphAny-level complexity by simply narrowing the search space, or removing it entirely and using a narrow set of fixed operators. The most important factor is that GOBLIN is *flexible* and can adapt its computational budget to the level of robustness required — a low-exploration GOBLIN basis essentially reduces to GraphAny. GraphAny, by construction, cannot do this.

## E  MEASURING RANGE FOR LINEAR GNN MoE MODELS

### E.1  WHY WE USE FIXED-$W$ RANGE

A linear GNN expert has the form

$$\hat{Y} = SXW, \qquad W = (SX)_L^+ Y_L,$$

so its Jacobian decomposes as

$$\frac{\partial \hat{Y}}{\partial X} = \frac{\partial (SX)}{\partial X} W \; + \; (SX)\frac{\partial W}{\partial X}.$$

The first term captures *explicit architectural mixing*: how information is propagated across the graph by the operator $S$. The second term arises solely from refitting $W$ to the labeled nodes and reflects sensitivity of the *regression solution*, not additional message passing.

Crucially, this second term does not introduce new graph-dependent paths by which information can propagate between nodes. Any dependence of $\hat{Y}_u$ on $X_v$ induced by $\partial W/\partial X$ must already be mediated through $SX$, and therefore cannot extend the receptive field beyond that encoded by $S$. As a result, it cannot increase the effective range of the model.

From an architectural perspective, $S$ fully specifies the computation graph through which node features interact; $W$ merely selects a linear readout within that fixed interaction structure. Since our goal is to diagnose *architectural under-reaching*, isolating the contribution of $S$ is sufficient and appropriate.

This mirrors standard practice in range and receptive-field analyses of trained GNNs, where weights are treated as fixed and range is understood as a property of the message-passing structure (Bamberger et al., 2025).

We also find this black-box range to be uninformative in practice; below we compute both types of range for GraphAny's linear GNN operators on several benchmark tasks and detail the relative weights for checkpointed GraphAny instances.

### E.2  FIXED-$W$ AND BLACK BOX RANGES FOR LINEAR GNNs FOR BENCHMARK GRAPHS

This section contains tables for 15 datasets; all *available, connected datasets* used by Zhao et al. (2024), except for the three largest ones, for which it is computationally infeasible to obtain the all-pairs geodesic distance. Let $\rho_{\overline{W}}$ denote fixed-weight linear GNN range as it is derived in the paper, and $\rho_{\text{B}}$ denote black-box range computed with `torch.autograd.functional.jacobian` for the end-to-end linear GNN solve. These results are summarized in Table 8, with results for individual datasets given in Tables 9–25.

**Discussion.**  $\rho_{\overline{W}}$ is around 0.6–1.0 for all datasets, usually about 0.9. The split between linear GNNs is relatively even everywhere; almost every weight is close to 0.33, and none are below 0.22 or above 0.48.

Table 8: Overall ranges (and for reference, validation accuracies), both fixed-$W$ and black box, for 17 connected benchmark datasets. Computation of $\rho_{\mathrm{B}}$ uses 500 sampled nodes, or all of them, whichever is fewer. In most cases, $\rho_B$ tracks mean geodesic distance fairly closely; $\rho_{\bar{W}}$ provides a more accurate version of range—due to the fixed $\rho \leq 2$ operators used, effective range of GraphAny on all tasks is $< 1$. All ranges are computed on the checkpointed instantiation of GraphAny from Zhao et al. (2024), trained on Wisconsin.

| Dataset | $\rho_{\bar{W}}$ | $\rho_{\mathrm{B}}$ | Mean geodesic distance | Val. Acc. (%) |
|---|---|---|---|---|
| AirBrazil | 0.80 | 1.63 | 2.17 | 50.00 |
| Cornell | 0.80 | 3.22 | 3.18 | 71.19 |
| Texas | 0.80 | 2.79 | 3.02 | 74.14 |
| AirEU | 0.82 | 1.29 | 2.32 | 34.59 |
| Squirrel | 0.91 | 2.34 | 3.09 | 42.31 |
| Wisconsin | 0.80 | 3.18 | 3.24 | 62.50 |
| BlogCatalog | 0.88 | 1.84 | 2.51 | 76.04 |
| Chameleon | 0.89 | 2.42 | 3.56 | 62.14 |
| Actor | 0.92 | 2.35 | 4.11 | 30.26 |
| Minesweeper | 0.81 | 29.87 | 46.66 | 80.16 |
| Tolokers | 0.93 | 2.12 | 2.79 | 78.26 |
| CoCS | 0.82 | 3.65 | 5.43 | 91.09 |
| Pubmed | 0.89 | 5.78 | 6.34 | 77.60 |
| Roman | 0.71 | 2152.7 | 2331.5 | 63.57 |
| AmzRatings | 0.80 | 16.04 | 16.24 | 42.05 |
| CoPhysics | 0.82 | 2.84 | 5.16 | 92.38 |
| Questions | 0.95 | 3.62 | 4.29 | 97.08 |

For most graphs variation in $\rho_{\mathrm{B}}$ between different linear GNNs (except for $X$, which is typically much lower) is minimal; some amount of mixing appears to be significant, but there is little difference between 1- and 2-hop. The second power is usually a little higher though. Overall $\rho_{\mathrm{B}}$ values do not vary much between *training* datasets on the same *evaluation* dataset, possibly because all of the training datasets are essentially short-range. There is some variation across evaluation datasets in $\rho_{\mathrm{B}}$. In general we see that the range skews fairly close to the mean_dist, i.e. the average SPD between node pairs. Given that the linear GNN features are analytic solutions that use the subset of labeled nodes, it is likely that the solutions depend heavily on labeled nodes. If we assume that some labeled node $w$ is approximately as close to some node $u$ as any other non-labeled node $v$, it makes sense that the range would track this value. Roman is particularly unusual — because it is close to a chain graph, with low degree and a very large diameter, its range ends up being extremely high. Minesweeper is similar.

Table 9: Ranges and weights of GraphAny linear GNN components for the **AirBrazil** dataset.

| LinearGNN | $\rho_{\bar{W}}$ | $\rho_B$ (131) | $\alpha_{\text{Cora}}$ | $\alpha_{\text{Wisconsin}}$ | $\alpha_{\text{Arxiv}}$ | $\alpha_{\text{Product}}$ |
|---|---|---|---|---|---|---|
| $X$ | 0.00 | 1.36 | 0.30 | 0.37 | 0.43 | 0.37 |
| $AX$ | 1.00 | 1.75 | 0.35 | 0.35 | 0.34 | 0.35 |
| $A^2X$ | 1.61 | 1.84 | 0.35 | 0.28 | 0.23 | 0.28 |
| $(I-A)X$ | 0.50 | 1.84 | — | — | — | — |
| $(I-A)^2X$ | 0.87 | 1.77 | — | — | — | — |

Table 10: Ranges and weights of GraphAny linear GNN components for the **Cornell** dataset.

| LinearGNN | $\rho_{\bar{W}}$ | $\rho_B$ (183) | $\alpha_{\text{Cora}}$ | $\alpha_{\text{Wisconsin}}$ | $\alpha_{\text{Arxiv}}$ | $\alpha_{\text{Product}}$ |
|---|---|---|---|---|---|---|
| $X$ | 0.00 | 2.98 | 0.30 | 0.34 | 0.35 | 0.37 |
| $AX$ | 1.00 | 3.42 | 0.35 | 0.38 | 0.36 | 0.34 |
| $A^2X$ | 1.48 | 3.23 | 0.35 | 0.29 | 0.30 | 0.29 |
| $(I-A)X$ | 0.50 | 3.09 | — | — | — | — |
| $(I-A)^2X$ | 0.86 | 3.18 | — | — | — | — |

Table 11: Ranges and weights of GraphAny linear components for the **Texas** dataset.

| LinearGNN | $\rho_{\bar{W}}$ | $\rho_B$ (183) | $\alpha_{\text{Cora}}$ | $\alpha_{\text{Wisconsin}}$ | $\alpha_{\text{Arxiv}}$ | $\alpha_{\text{Product}}$ |
|---|---|---|---|---|---|---|
| $X$ | 0.00 | 2.82 | 0.30 | 0.32 | 0.33 | 0.34 |
| $AX$ | 1.00 | 2.11 | 0.35 | 0.42 | 0.36 | 0.34 |
| $A^2X$ | 1.50 | 3.89 | 0.35 | 0.25 | 0.31 | 0.31 |
| $(I-A)X$ | 0.50 | 2.94 | — | — | — | — |
| $(I-A)^2X$ | 0.87 | 3.02 | — | — | — | — |

Table 12: Ranges and weights of GraphAny linear GNN components for the **Wisconsin** dataset.

| LinearGNN | $\rho_{\bar{W}}$ | $\rho_B$ (251) | $\alpha_{\text{Cora}}$ | $\alpha_{\text{Wisconsin}}$ | $\alpha_{\text{Arxiv}}$ | $\alpha_{\text{Product}}$ |
|---|---|---|---|---|---|---|
| $X$ | 0.00 | 2.94 | 0.30 | 0.34 | 0.34 | 0.36 |
| $AX$ | 1.00 | 3.28 | 0.35 | 0.38 | 0.36 | 0.34 |
| $A^2X$ | 1.48 | 3.32 | 0.35 | 0.29 | 0.30 | 0.30 |
| $(I-A)X$ | 0.50 | 3.06 | — | — | — | — |
| $(I-A)^2X$ | 0.86 | 3.11 | — | — | — | — |

Table 13: Ranges and weights of GraphAny linear components for the **AirEU** dataset.

| LinearGNN | $\rho_{\bar{W}}$ | $\rho_B$ (399) | $\alpha_{\text{Cora}}$ | $\alpha_{\text{Wisconsin}}$ | $\alpha_{\text{Arxiv}}$ | $\alpha_{\text{Product}}$ |
|---|---|---|---|---|---|---|
| $X$ | 0.00 | 0.36 | 0.29 | 0.40 | 0.42 | 0.48 |
| $AX$ | 1.00 | 1.68 | 0.35 | 0.29 | 0.33 | 0.30 |
| $A^2X$ | 1.71 | 2.11 | 0.36 | 0.31 | 0.25 | 0.22 |
| $(I-A)X$ | 0.50 | 1.90 | — | — | — | — |
| $(I-A)^2X$ | 0.91 | 2.29 | — | — | — | — |

Table 14: Ranges and weights of GraphAny linear components for the **Squirrel** dataset.

| LinearGNN | $\rho_{\bar{W}}$ | $\rho_B$ (500) | $\alpha_{\text{Cora}}$ | $\alpha_{\text{Wisconsin}}$ | $\alpha_{\text{Arxiv}}$ | $\alpha_{\text{Product}}$ |
|---|---|---|---|---|---|---|
| $X$ | 0.00 | 1.44 | 0.30 | 0.35 | 0.34 | 0.38 |
| $AX$ | 1.00 | 2.64 | 0.35 | 0.31 | 0.33 | 0.31 |
| $A^2X$ | 1.76 | 2.99 | 0.35 | 0.34 | 0.33 | 0.31 |
| $(I-A)X$ | 0.50 | 2.38 | — | — | — | — |
| $(I-A)^2X$ | 0.93 | 2.59 | — | — | — | — |

Table 15: Ranges and weights of GraphAny linear components for the **BlogCatalog** dataset.

| LinearGNN | $\rho_{\bar{W}}$ | $\rho_B$ (500) | $\alpha_{\text{Cora}}$ | $\alpha_{\text{Wisconsin}}$ | $\alpha_{\text{Arxiv}}$ | $\alpha_{\text{Product}}$ |
|---|---|---|---|---|---|---|
| $X$ | 0.00 | 0.92 | 0.30 | 0.38 | 0.37 | 0.41 |
| $AX$ | 1.00 | 2.38 | 0.35 | 0.33 | 0.34 | 0.33 |
| $A^2X$ | 1.89 | 2.42 | 0.35 | 0.29 | 0.29 | 0.26 |
| $(I-A)X$ | 0.50 | 2.37 | — | — | — | — |
| $(I-A)^2X$ | 0.97 | 2.37 | — | — | — | — |

Table 16: Ranges and weights of GraphAny linear components for the **Chameleon** dataset.

| LinearGNN | $\rho_{\bar{W}}$ | $\rho_B$ (500) | $\alpha_{\text{Cora}}$ | $\alpha_{\text{Wisconsin}}$ | $\alpha_{\text{Arxiv}}$ | $\alpha_{\text{Product}}$ |
|---|---|---|---|---|---|---|
| $X$ | 0.00 | 1.13 | 0.30 | 0.31 | 0.30 | 0.32 |
| $AX$ | 1.00 | 2.79 | 0.35 | 0.40 | 0.35 | 0.34 |
| $A^2X$ | 1.70 | 3.33 | 0.35 | 0.29 | 0.35 | 0.34 |
| $(I-A)X$ | 0.50 | 2.47 | — | — | — | — |
| $(I-A)^2X$ | 0.90 | 2.74 | — | — | — | — |

Table 17: Ranges and weights of GraphAny linear components for the **Actor** dataset.

| LinearGNN | $\rho_{\bar{W}}$ | $\rho_B$ (500) | $\alpha_{\text{Cora}}$ | $\alpha_{\text{Wisconsin}}$ | $\alpha_{\text{Arxiv}}$ | $\alpha_{\text{Product}}$ |
|---|---|---|---|---|---|---|
| $X$ | 0.00 | 1.00 | 0.30 | 0.33 | 0.35 | 0.38 |
| $AX$ | 1.00 | 2.39 | 0.35 | 0.30 | 0.33 | 0.31 |
| $A^2X$ | 1.69 | 3.55 | 0.35 | 0.36 | 0.33 | 0.30 |
| $(I-A)X$ | 0.50 | 2.46 | — | — | — | — |
| $(I-A)^2X$ | 0.92 | 3.11 | — | — | — | — |

Table 18: Ranges and weights of GraphAny linear components for the **Minesweeper** dataset.

| LinearGNN | $\rho_{\bar{W}}$ | $\rho_B$ (500) | $\alpha_{\text{Cora}}$ | $\alpha_{\text{Wisconsin}}$ | $\alpha_{\text{Arxiv}}$ | $\alpha_{\text{Product}}$ |
|---|---|---|---|---|---|---|
| $X$ | 0.00 | 26.63 | 0.31 | 0.32 | 0.32 | 0.35 |
| $AX$ | 1.00 | 30.65 | 0.36 | 0.34 | 0.34 | 0.33 |
| $A^2X$ | 1.37 | 32.10 | 0.34 | 0.34 | 0.35 | 0.32 |
| $(I-A)X$ | 0.50 | 34.24 | — | — | — | — |
| $(I-A)^2X$ | 0.81 | 30.59 | — | — | — | — |

Table 19: Ranges and weights of GraphAny linear components for the **Tolokers** dataset.

| LinearGNN | $\rho_{\bar{W}}$ | $\rho_B$ (500) | $\alpha_{\text{Cora}}$ | $\alpha_{\text{Wisconsin}}$ | $\alpha_{\text{Arxiv}}$ | $\alpha_{\text{Product}}$ |
|---|---|---|---|---|---|---|
| $X$ | 0.00 | 2.02 | 0.30 | 0.33 | 0.32 | 0.35 |
| $AX$ | 1.00 | 1.98 | 0.35 | 0.36 | 0.35 | 0.34 |
| $A^2X$ | 1.82 | 2.39 | 0.34 | 0.32 | 0.32 | 0.30 |
| $(I-A)X$ | 0.50 | 2.05 | — | — | — | — |
| $(I-A)^2X$ | 0.95 | 2.05 | — | — | — | — |

Table 20: Ranges and weights of GraphAny linear components for the **CoCS** dataset.

| LinearGNN | $\rho_{\bar{W}}$ | $\rho_B$ (500) | $\alpha_{\text{Cora}}$ | $\alpha_{\text{Wisconsin}}$ | $\alpha_{\text{Arxiv}}$ | $\alpha_{\text{Product}}$ |
|---|---|---|---|---|---|---|
| $X$ | 0.00 | 2.21 | 0.30 | 0.34 | 0.31 | 0.34 |
| $AX$ | 1.00 | 4.14 | 0.35 | 0.37 | 0.36 | 0.35 |
| $A^2X$ | 1.54 | 4.73 | 0.34 | 0.29 | 0.33 | 0.31 |
| $(I-A)X$ | 0.50 | 4.02 | — | — | — | — |
| $(I-A)^2X$ | 0.87 | 4.41 | — | — | — | — |

Table 21: Ranges and weights of GraphAny linear components for the **Pubmed** dataset.

| LinearGNN | $\rho_{\overline{W}}$ | $\rho_B$ (500) | $\alpha_{\text{Cora}}$ | $\alpha_{\text{Wisconsin}}$ | $\alpha_{\text{Arxiv}}$ | $\alpha_{\text{Product}}$ |
|---|---|---|---|---|---|---|
| $X$ | 0.00 | 6.02 | 0.30 | 0.34 | 0.35 | 0.39 |
| $AX$ | 1.00 | 5.63 | 0.35 | 0.34 | 0.35 | 0.33 |
| $A^2X$ | 1.71 | 5.69 | 0.34 | 0.32 | 0.30 | 0.28 |
| $(I-A)X$ | 0.50 | 5.83 | — | — | — | — |
| $(I-A)^2X$ | 0.93 | 5.73 | — | — | — | — |

Table 22: Ranges and weights of GraphAny linear components for the **Roman** dataset.

| LinearGNN | $\rho_{\overline{W}}$ | $\rho_B$ (500) | $\alpha_{\text{Cora}}$ | $\alpha_{\text{Wisconsin}}$ | $\alpha_{\text{Arxiv}}$ | $\alpha_{\text{Product}}$ |
|---|---|---|---|---|---|---|
| $X$ | 0.00 | 2088.70 | 0.30 | 0.33 | 0.35 | 0.39 |
| $AX$ | 1.00 | 2212.81 | 0.35 | 0.26 | 0.29 | 0.28 |
| $A^2X$ | 1.12 | 2166.40 | 0.35 | 0.40 | 0.36 | 0.34 |
| $(I-A)X$ | 0.50 | 2173.21 | — | — | — | — |
| $(I-A)^2X$ | 0.75 | 2179.69 | — | — | — | — |

Table 23: Ranges and weights of GraphAny linear components for the **AmzRatings** dataset.

| LinearGNN | $\rho_{\overline{W}}$ | $\rho_B$ (500) | $\alpha_{\text{Cora}}$ | $\alpha_{\text{Wisconsin}}$ | $\alpha_{\text{Arxiv}}$ | $\alpha_{\text{Product}}$ |
|---|---|---|---|---|---|---|
| $X$ | 0.00 | 16.10 | 0.31 | 0.32 | 0.31 | 0.35 |
| $AX$ | 1.00 | 15.98 | 0.35 | 0.35 | 0.35 | 0.34 |
| $A^2X$ | 1.39 | 16.06 | 0.34 | 0.33 | 0.34 | 0.32 |
| $(I-A)X$ | 0.50 | 15.99 | — | — | — | — |
| $(I-A)^2X$ | 0.80 | 15.81 | — | — | — | — |

Table 24: Ranges and weights of GraphAny linear components for the **CoPhysics** dataset.

| LinearGNN | $\rho_{\overline{W}}$ | $\rho_B$ (500) | $\alpha_{\text{Cora}}$ | $\alpha_{\text{Wisconsin}}$ | $\alpha_{\text{Arxiv}}$ | $\alpha_{\text{Product}}$ |
|---|---|---|---|---|---|---|
| $X$ | 0.00 | 0.83 | 0.30 | 0.33 | 0.29 | 0.31 |
| $AX$ | 1.00 | 3.36 | 0.35 | 0.39 | 0.38 | 0.37 |
| $A^2X$ | 1.57 | 4.54 | 0.34 | 0.27 | 0.34 | 0.32 |
| $(I-A)X$ | 0.50 | 3.05 | — | — | — | — |
| $(I-A)^2X$ | 0.88 | 4.10 | — | — | — | — |

Table 25: Ranges and weights of GraphAny linear components for the **Questions** dataset.

| LinearGNN | $\rho_{\overline{W}}$ | $\rho_B$ (200) | $\alpha_{\text{Cora}}$ | $\alpha_{\text{Wisconsin}}$ | $\alpha_{\text{Arxiv}}$ | $\alpha_{\text{Product}}$ |
|---|---|---|---|---|---|---|
| $X$ | 0.00 | 3.34 | 0.30 | 0.32 | 0.32 | 0.35 |
| $AX$ | 1.00 | 3.46 | 0.35 | 0.35 | 0.35 | 0.34 |
| $A^2X$ | 1.82 | 4.07 | 0.34 | 0.33 | 0.33 | 0.31 |
| $(I-A)X$ | 0.50 | 3.84 | — | — | — | — |
| $(I-A)^2X$ | 0.96 | 3.82 | — | — | — | — |

