# OpenReview forum: "Can Graph Foundation Models Generalize Over Architecture?"
_ICLR.cc/2026/Workshop/GRaM — ICLR 2026 Workshop GRaM Poster_

### Official Review · Reviewer_mFHF · 2026-02-17
**Graph Foundation Models**

**Rating:** 6
**Confidence:** 3

**Review:**

The paper argues that current domain-agnostic graph foundation models (e.g., GraphAny, triple-symmetry GNNs) implicitly assume a fixed message-passing architecture and therefore fail under architectural mismatch across tasks. It formalizes this via task/model range and introduces a controlled synthetic task (kHopSign) where required range varies with k.

To address this, it proposes GOBLIN: an inference-time framework that searches for and mixes task-specific linear graph operators (Gaussian distance-localized and heat-kernel diffusion families), using analytic least-squares solves on labeled nodes and a permutation-invariant DeepSet-style mixer.

The paper is interesting and can be recommended for acceptance, but there are also some weaknesses:

- Zero-shot is somewhat nuanced; GOBLIN performs per-graph label-driven operator search/tuning at inference (even if training-free / analytic), which should be discussed explicitly and compared fairly to baseline tuning allowances.

- Inference-time cost/scaling: BO/search + multiple solves + distance/diffusion computations may be heavy; runtime numbers would help.

- Ablations are missing, and contribution of (i) operator families, (ii) search (budget/priors), (iii) DeepSet mixing is not isolated.

My evaluation is therefore 6.

**Pmlr Suitability:**

Yes

---

### Official Review · Reviewer_HWbC · 2026-02-21
**Architectural adaptivity is interesting, but claims are stronger than evaluation**

**Rating:** 6
**Confidence:** 3

**Review:**

This paper argues that existing domain-agnostic GFMs are limited by fixed architectural backbones that constrain their effective receptive field. The authors formalize this via range, a Jacobian-based diagnostic, prove that fixed operator bases impose hard ceilings on task solvability, and introduce GOBLIN, a framework that discovers task-appropriate linear graph operators at inference time via Bayesian optimization.

### Strengths

* The distinction between domain generalization and architectural generalization is interesting. The observation that standard GFM benchmarks are all short-range tasks, masking a structural limitation, is also an interesting point.

* The two-stage pipeline (BO-based operator discovery + Deep Set weighting) is modular and naturally motivated by the identified limitation.

### Weaknesses

* The central claim is about architectural rigidity in GFMs broadly, but the only baselines are two shallow MPNN-based methods. Graph transformers with global attention provide implicitly variable-range receptive fields. Exphormer proves O(log n) layers suffice for all-pairs interaction, GraphGPS would be another baseline to include. Without such baselines, it is unclear whether the range limitation is fundamental to GFMs or specific to the two methods studied.

* GOBLIN's operator family is closely related to learnable spectral filters such as GPR-GNN, which adaptively learns polynomial filters over the graph spectrum. The paper should clarify its novelty relative to this line of work.

* The ~2% average improvement is driven by three heterophilous datasets (Chameleon, Squirrel, AirBrazil), while GOBLIN underperforms on several others (Questions, Wiki, Cornell). The gains may stem from the operator family implicitly learning better filters for heterophily rather than from range adaptation. The paper does not disentangle these explanations.

* GOBLIN requires geodesic distance computation, ~15 BO evaluations with linear solves, and mixture-of-experts training, which is substantially heavier than GraphAny's 5 analytic solves. No wall-clock comparisons or scaling analysis are provided.

* The paper claims architecture adaptivity is necessary for "true GFMs" but validates along one axis (range) on one task type (transductive node classification), trains the mixture-of-experts on Cora alone with no sensitivity analysis, and averages over only 4 seeds with high variance on several datasets.

**Pmlr Suitability:**

No

---

### Official Review · Reviewer_nKjc · 2026-02-24
**Generalization over architectures**

**Rating:** 6
**Confidence:** 3

**Review:**

This paper studies whether current GFMs truly generalize across architectures. They focus on GraphAny and TSGNN and argue that their fixed (short-range) backbones may fail to generalize to tasks requiring broader receptive fields. The authors formalize this limitation through a range-based diagnostic and propose GOBLIN, a framework that adapt the architecture on the target graph at inference-time. Moreover, they introduce a synthetic benchmark (kHopSign) to illustrate the shortcomings of fixed-range approaches as task range increases.

Strengths:
- The disentanglement between domain generalization and architectural generalization is interesting and under-explored
- The original synthetic dataset
- Clear diagnostic via range

Limitations:
- While motivated, range is the only architectural axis considered, which makes the their analysis somewhat preliminary. Please, comment on some real-word datasets (e.g., AirEU, WkCS) requiring range $\sim 2.5$ where GraphAny outperforms GOBLIN, suggesting that other factors may influence GFMs ability to generalize.
- Comparison limited to two GFMs based on MPNNs with depth <=2 layers but in practice limited to GraphAny.
- Several design choices appear heuristic and insufficiently motivated, such as the selection of two objective-maximizing sampled operators as a basis, along with the fixed inclusion of $A^2$, and the rationale for selecting Cora as the pretraining graph.
- The claim that other methods are within 0.5% of each other and that GOBLIN is about 2% holds solely when considering the averaged accuracy across datasets. The improvement is driven by a subset of benchmarks with large gains (e.g., Chameleon, Squirrel), while performance differences are small or even reversed on several other datasets.

Minor modifications:
- Please, add the explicit equation of $J_{uv}$ in terms of $S_{uv}$ and $W_{d'c}$ in the main text (around line 197) to enhance readability
- Eq. 3 at line 227 references the wrong equation
- The color highlighting in Table 2 (green/orange) appears inconsistent with a “best/second-best per row” interpretation. For example, in AirUS the second-best value is not highlighted, and in AmzComp and AmzPhoto the second-best values should be GOBLIN's. Please clarify the coloring rule and correct potential formatting errors.

While the paper appears somewhat preliminary, it addresses an important and promising direction, and, in my opinion, it fits the workshop.

**Pmlr Suitability:**

Yes

---

### Official Review · Reviewer_poKt · 2026-02-24
**A promising idea but the current experiments are too limited**

**Rating:** 5
**Confidence:** 4

**Review:**

This paper raises a valid and interesting criticism of current graph foundation models—namely, that fixed-backbone architectures may fail when task-specific architectural requirements (especially range) differ—and the GOBLIN framework is a thoughtful attempt to address that via inference-time operator selection, but the paper still feels too preliminary and overclaims relative to the evidence. The central thesis (“true GFMs need architecture adaptivity”) is plausible, yet the validation leans heavily on a synthetic range-controlled task and a zero-shot node-classification setup inherited from prior work, which limits how broadly the conclusions should be interpreted. The reported benchmark gains are encouraging (average improvement on the 25-dataset suite), but they appear uneven and are concentrated in a few datasets, while GOBLIN also underperforms on several others; this makes the robustness narrative less convincing than the paper suggests. In addition, the method introduces extra inference-time complexity and heuristic design choices (operator family, BO search budget, feature summaries, fixed operator injection), and it is not yet clear how sensitive results are to these choices or how well the approach scales to larger graphs where shortest-path-based operators become costly. Overall, this reads like a promising workshop paper with a strong idea and good diagnostics, but not yet a fully convincing demonstration of a broadly applicable graph foundation model advance.

**Pmlr Suitability:**

No

---

### Meta-Review · Area_Chair_1oWQ · 2026-02-26

**Decision:**

Accept

**Metareview:**

This paper tackles an important and timely problem in graph learning. The proposed method is sound. However, reviewers have raised that claims made are stronger than evidence suggests and that baselines are too limited. Authors are encouraged to address these points.

**Relevance To Proceedings:**

Yes — suitable for PMLR (long paper)

**Relevance To Workshop:**

Yes — suitable for GRaM

---

### Decision · Program_Chairs · 2026-03-02

Accept (Poster)